# Semantic relatedness proactively benefits learning, memory, and interdependence across episodes

Kelly A Bennion[1]*, Jade Phong[1], Mytien Le[1], Kunhua Cheng[1], Christopher N Wahlheim[2], James W Antony[1]*

[1]Department of Psychology and Child Development, California Polytechnic State University, San Luis Obispo, United States; [2]Department of Psychology, University of North Carolina at Greensboro, Greensboro, United States

*For correspondence:
kbennion@calpoly.edu (KAB);
jwantony@calpoly.edu (JWA)

Competing interest: The authors declare that no competing interests exist.

## eLife Assessment

This **important** work advances our understanding of how memories interact to facilitate or interfere with one another, also informing our understanding of how humans build knowledge. The study provides **compelling** evidence that semantic relatedness proactively benefits memory using clean experimental design, rigorous statistics, large N samples, and well-characterized stimuli. The study also demonstrates the boundaries of these proactive benefits, showing that when studied items have weaker semantic relationships, proactive interference may be observed. This research will be of interest to memory researchers as well as cognitive psychologists, neuroscientists, and educators more broadly.

**Abstract** Over the past century of memory research, the interplay between initial and later-learned information in determining long-term memory retention has been of central interest. A likely factor for determining whether initial and later memories interfere with or strengthen each other is semantic relatedness. Relatedness has been shown to *retroactively* boost initial memory and increase the interdependence between earlier and more recent experiences in memory. Here, we investigated the converse relationship of how relatedness *proactively* affects later memory for paired associates. In five experiments (N=1000 total), we varied the relatedness between initial and later cues, initial and later targets, or both. Across experiments and conditions, relatedness profoundly benefited later-learned memories – in some conditions, low relatedness reliably produced proactive interference (versus a control condition) while high relatedness produced proactive facilitation within the same experiment. Additionally, relatedness also accelerated learning and increased interdependence between initial and later-learned pairs. In sum, we demonstrate the robust effects of relatedness in scaffolding memory for recently learned information and creating strong integrative links with prior experiences.

## Introduction

When one begins learning a set of associations in a new domain, the associations do not exist in a vacuum but rather affect each other in various ways. Consider the example that a novice bartender learns that a 'martini' is made with *dry vermouth* and later that a 'Manhattan' is made with *sweet vermouth*. In this case, previously learning about the martini could affect memory for the Manhattan by diminishing it (proactive interference [PI]) or improving it (proactive facilitation [PF]).

Here, we ask whether this outcome can be predicted by the *semantic relatedness* between the learning events. In so doing, we re-visit a classic proposal by *Osgood, 1949* that has been elaborated upon in the intervening decades (*Bugelski and Cadwallader, 1956*; *Eich, 1982*; *Mensink and Raaijmakers, 1988*) that memory changes critically depend on the relatedness between events. A recent paper of ours (*Antony et al., 2022b*) showed that later learning affected memory for prior events (i.e. retroactively) based on relatedness. The current paper can be considered a twin companion of that prior effort, in that we are keeping the same subject population, stimuli, experimental procedure, and timing, and we vary only the order of list learning to focus on proactive, rather than retroactive, effects. Additionally, we mirror that paper's formatting and figures to ease comparisons among the studies, and we directly compare some data between the studies.

There are theoretical reasons to support directly opposing outcomes for how semantic relatedness with previously learned stimuli proactively affects new episodic memories (*Brunmair and Richter, 2019*). That is, relatedness could interfere with the new memory, promoting PI, or it could cause one to recall the prior memory and link it with the new one, which may make the new memory more resistant to forgetting and promote PF. In favor of the PI account, the presence of multiple targets for one cue could benefit from more mental separation among the targets (*Underwood, 1969*). In fact, increasing relatedness in some paradigms can increase interference (*Bower et al., 1994*; *McGeoch and McDonald, 1931*; *McGeoch and McGeoch, 1937*) and/or intrusions (*Dallett, 1962*; *Dallett, 1964*; *Osgood, 1946*; *Postman, 1961*; *Underwood, 1951*), and intrusions have been shown to increase specifically with similarity in proactive paradigms (*Young, 1955*). Conversely, in favor of the PF account, we consider two main related theories. The first is the importance of 'remindings' in memory, which involve reinstating representations from an earlier study phase during later learning (*Hintzman, 2011*). This idea centers study-phase retrieval, which involves being able to mentally recall prior information and is usually applied to exact repetitions of the same material (*Benjamin and Tullis, 2010*; *Hintzman et al., 1975*; *Siegel and Kahana, 2014*; *Thios and D'Agostino, 1976*; *Zou et al., 2023*). However, remindings can occur upon the presentation of related (but not identical) material and can result in better memory for both prior and new information when memory for the linked events becomes more interdependent (*Hintzman, 2011*; *Hintzman et al., 1975*; *McKinley et al., 2019*; *McKinley and Benjamin, 2020*; *Schlichting and Preston, 2017*; *Tullis et al., 2014*; *Wahlheim and Zacks, 2019*). The second is the memory-for-change framework, which builds upon these ideas and argues that humans often retrieve prior experiences during new learning, either spontaneously by noticing changes from what was learned previously or by instruction (*Jacoby et al., 2015*; *Jacoby and Wahlheim, 2013*). The key advance of this framework is that recollecting changes is necessary for PF, whereas PI occurs without recollection. This framework has been applied to paradigms including stimulus changes, including common paired associate paradigms (e.g. A-B, A-D) that we cover extensively later. Because humans may be more likely to notice and recall prior information when it is more related to new information, these two accounts would predict that semantic relatedness instead promotes successful remindings, which would create PF and interdependence among the traces.

One popular method for testing these accounts is to use paired associated learning paradigms in which original associations (A-B) change on a later presentation (A-D; *Barnes and Underwood, 1959*; *Briggs, 1954*). Here, we refer to this A-B, A-D paradigm as ΔTarget learning. Under conditions of no semantic relatedness between the B and D targets, ΔTarget learning typically causes PI in the form of impaired A-D memory relative to a control condition, likely because B and D compete at retrieval (*Anderson and Neely, 1996*; *Bower et al., 1994*; *Caplan et al., 2014*; *Castro et al., 2002*; *Jung, 1962*; *Jung, 1967*; *Postman, 1962*; *Postman and Underwood, 1973*; *Twedt and Underwood, 1959*; *Young, 1955*). However, as the semantic relationship between B and D increases, PI decreases and, in some cases, PF occurs (*Bruce, 1933*; *Dallett, 1964*; *Martin and Dean, 1964*; *Mehler and Miller, 1964*; *Metcalfe et al., 1993*; *Morgan and Underwood, 1950*; *Osgood, 1946*; *Russell and Storms, 1955*; although see *Mills, 1973* for one case in which PI increases) An important early result using ΔTarget learning showed PF only when participants performed a mental linking strategy, in which they thought back to and integrated the prior and new pairs (*Martin and Dean, 1964*). Collectively, these findings indicate that prior learning can improve or impair memory for more recent information based on how people notice and consider relationships to prior information during new learning.

Over the last decade, several studies have rigorously addressed this issue by asking participants during new learning if they notice changes in new stimuli relative to existing memories, and if they

can recall the existing memories. On later memory tests of the updated associations, participants are also asked to indicate if they recollect earlier changes. These studies consistently showed that noticing and recollecting changes are associated with enhanced memory for new associations (for a review, see *Wahlheim et al., 2021*). Largely, these proactive benefits parallel those in the retroactive direction – that is, increased similarity between B and D also reduces retroactive interference (RI) or causes retroactive facilitation (RF; *Antony et al., 2022b*; *Barnes and Underwood, 1959*; *Dallett, 1962*; *Kanungo, 1967*; *Osgood, 1946*; *Young, 1955*); we review the similarities and differences between proactive and retroactive operations more extensively below. In sum, proactive benefits of relatedness suggest that higher semantic relatedness may result in A-B and A-D becoming integrated (*Wahlheim, 2014*), perhaps by promoting the extent to which changed associations trigger remindings of existing memories with both shared and unique features.

An association can be modified in ways other than changing the target. Indeed, in Osgood's proposal and in our prior paper (*Antony et al., 2022b*), we investigated how numerous associational changes affected original memories. Under A-B, C-B learning conditions, which we call ΔCue learning, C-B memory relative to memory for unrelated stimuli (e.g. C-D) tends to be similar (recall condition of *Postman et al., 1969*; *Traxler, 1973*) or slightly worse (*Amundson et al., 2003*; *Postman et al., 1969*; *Twedt and Underwood, 1959*; see *Amundson et al., 2003* for a similar finding in rodents). Nevertheless, new learning of C-B associations can occur more quickly with ΔCue learning (*Mandler and Heinemann, 1956*), and it appears at least possible that under certain conditions, C-B could also show proactive facilitation in subsequent memory, perhaps by making B responses more accessible (*Estes, 1979*; *Martin, 1965*). When instead investigating retroactive effects of ΔCue learning, studies have found little-to-no RI when A and C are unrelated (*Houston, 1966*; *Keppel and Underwood, 1962*; *Twedt and Underwood, 1959*) and even RF when A and C are semantically related (*Antony et al., 2022b*; *Bugelski and Cadwallader, 1956*; *Hamilton, 1943*).

Finally, when learning associations with completely new cues and targets (typically called A-B, C-D learning, but which we call ΔBoth learning), the C-D memory is generally regarded as distinct from A-B. In the retroactive direction, when A is related to C and B related to D, RI has been observed when testing memory after a few minutes (*Baddeley and Dale, 1966*; *Bugelski and Cadwallader, 1956*; *McGeoch and McGeoch, 1937*; *Saltz and Hamilton, 1967*). However, when both relationships are very strong (e.g. learning beer-late followed by keg-tardy) and a longer delay is used, we previously showed reliable RF (*Antony et al., 2022b*). These latter results also accord with the idea that, with high similarity between both items of the pair, participants may think back and recollect prior associations, which aids their long-term maintenance. However, no studies to our knowledge have used ΔBoth learning with varying relationships between earlier and recent word pairs and tests of proactive effects of memory, as we do here.

To visualize this complicated collection of effects, Osgood proposed a three-dimensional surface by which old and new associations may interact to influence memory (*Eich, 1982*; *Mensink and Raaijmakers, 1988*; *Figure 1A*). Briefly, one can summarize Osgood's argument by carefully considering three continuous relationships between cues and targets. With a constant cue, one can draw a ΔTarget line from an unrelated to an identical target (i.e. from A-B, A-D to A-B, A-B). Somewhere along this line, PI should shift to PF. Next, with a constant target starting from the A-B, A-D point (i.e. from A-B, A-D to A-B, C-D), PI should disappear as the pair becomes a wholly new association. Finally, with a constant target starting from the A-B, A-B point towards A-B, C-B (the ΔCue line), Osgood posited that PI generally did not occur. Therefore, proactive effects went from no memory change at the A-B, C-B point to maximum PF at identity. Osgood also speculated how a full surface spanning all cue-target relatedness values would influence later memory (*Figure 1A*), although it was based on scant data.

Osgood largely considered proactive and retroactive effects of memory to be identical (or considered the data too insufficient to assert otherwise) – that is, he used the same surface to describe both effects. Nevertheless, there may be differences between them. First, if remindings retroactively benefit existing memories by evoking rehearsals (*Antony et al., 2017*; *Roediger and Karpicke, 2006*; *Rowland, 2014*) and promoting associations across episodes, then the conditions that promote remindings (such as strong relatedness or instructions to think back) could benefit existing memories more than new ones. That is, the additional retrieval trial would only occur for old, and not new, associations. Conversely, if the sole mechanism behind these benefits were the interdependence across

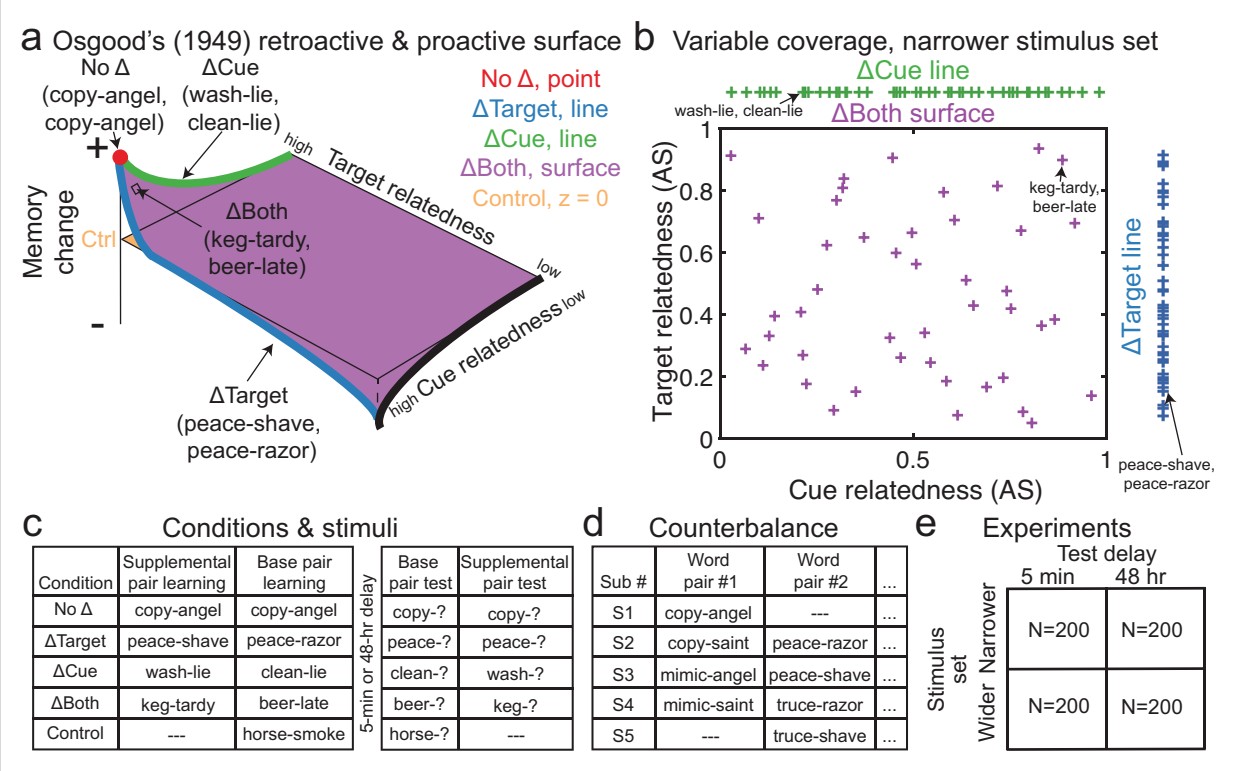

**Figure 1.** Overview of Osgood's predictions for proactive memory effects, distributions of relatedness values, conditions, stimuli, counterbalance procedures, and experiments. (**a**) We depict how our design would look if it confirmed Osgood's (1949) proposed surface. Cue and target relatedness are shown along the *y*- and *x*-axes, respectively, while memory change is shown along the *z*-axis, relative to the Control condition, which is simply depicted as the *z*=0 plane. (**b**) In the narrower stimulus set, variables covered the full range of associative strength (AS) values. Plus signs along the ΔCue (green) and ΔTarget (blue) lines depict how cue and target relatedness were respectively distributed. Purple plus signs inside the scatterplot show how cue and target relatedness values were distributed in the ΔBoth condition. (**c**) Pairs were divvied into four experimental conditions during supplemental pair learning (which was performed first), followed by base pair learning, which included a fifth Control condition. After 5 min or 48 hr, base and supplemental pairs were tested (in that order). (**d**) Each base pair was included in a different condition for every five subjects. (**e**) The primary four experiments involved crossing delays and stimulus sets. In (**a**) and (**b**), example word pairs from (**c**) are labeled for explanatory purposes. See also *Figure 1—figure supplement 1* for visualizations using the stimulus set with wider semantic relatedness.

The online version of this article includes the following figure supplement(s) for figure 1:

**Figure supplement 1.** Wider stimulus set examples and coverage.

episodes, then proactive and retroactive effects may be equal. Second, by virtue of being learned second and subject to possible fatigue or prior learning effects (*Postman and Keppel, 1977*; *Tulving and Watkins, 1974*; *Underwood, 1957*), newer associations may be more susceptible to interference than old associations when probed after a delay (but see *Melton and von Lackum, 1941* and *Szpunar et al., 2008* for how this interference is mitigated by intermittent testing, which we use here). If it were the case that interference occurred more strongly for the second-learned list, we would expect worse memory for even the ΔBoth or Control pairs in the proactive case than in the retroactive case.

In our prior paper, we examined how the full space of Osgood's predictions resulted in RI versus RF by systematically varying the semantic relatedness of cues, targets, or both. We did this using global vector similarity (GloVe; *Pennington et al., 2014*), a distributional semantic model, and associative strength (AS), collected using free association norms (*Nelson et al., 2004*), to obtain quasi-continuous measures of relatedness. We also obtained a large dataset (N=1000 across experiments) so that we could sample each point in the space under two different memory delays and two different average levels of relatedness. Here, we will use a nearly identical approach with a similar dataset to instead investigate *proactive* memory effects (N=800 across the first four experiments and N=200 for a final experiment performed for the purposes of comparing retroactive and proactive effects; *Figure 1B*).

Briefly, subjects initially learned 36 unrelated word pairs, which we will call *supplemental pairs* (***Figure 1C***). We call the first learned list supplemental pairs because they are less important to our aims – our primary memories of interest were the *base* pairs, which were learned afterwards. Supplemental pairs belonged to four of the five within-participant conditions: pairs in the No Δ condition were the same as later-learned base pairs (e.g. sick-push), whereas other pairs belonged to the ΔTarget (e.g. sick-shove), ΔCue (e.g. ill-push), or ΔBoth (e.g. ill-shove) conditions. Next, subjects learned 45 base pairs (e.g. sick-push), which also included pairs from a fifth (Control) condition that had no relationship to any supplemental pairs. In both blocks of learning, subjects learned each pair until they could correctly retrieve it once, after which it dropped out. Across every five subjects, pair assignments were counterbalanced, such that the same base pairs rotated among the five conditions (***Figure 1D***).

Across the first four experiments, we manipulated two factors: range of relatedness among the pairs and retention interval before the final test. The narrower range of relatedness used direct AS between pairs using free association norms, such that all pairs had between 0.03 and 0.96 association strength. Although this encompasses what appears to be a full range of relatedness values, pairs with even low AS are still related in the context of all possible associations (e.g. pious-holy has AS = 0.03 but would generally be considered related; ***Figure 1B***). The stimuli using a wider range of relatedness spanned the full range of global vector similarity (***Pennington et al., 2014***) that included many associations that would truly be considered unrelated (***Figure 1—figure supplement 1A***). One can see the range of the wider relatedness values in ***Figure 1—figure supplement 1B*** and comparisons between narrower and wider relatedness values in ***Figure 1—figure supplement 1C***. Numerous studies have shown that delays affect the extent of interference versus facilitation in retroactive and proactive designs (***Antony et al., 2022b***; ***Antony and Bennion, 2022a***; ***Baran et al., 2010***; ***Chan, 2009***; ***Liu and Ranganath, 2019***; ***Lustig et al., 2004***; ***Ortega et al., 2015***; ***Wahlheim et al., 2024***; ***Wixted, 2004***). Intriguingly, RI tends to decrease over time (***Antony et al., 2022b***; ***Antony and Bennion, 2022a***; ***Baran et al., 2010***; ***Chan, 2009***; ***Liu and Ranganath, 2019***; ***Lustig et al., 2004***; ***Ortega et al., 2015***; ***Wixted, 2004***), whereas PI tends to grow over time via increased intrusions and/or worse memory (***Greenberg and Underwood, 1950***; ***Mensink and Raaijmakers, 1988***; ***Postman et al., 1969***; ***Underwood, 1948***; ***Underwood, 1949***). We therefore employed two different delays of 5 min and 48 hr to assess how PI versus PF would change over time with varying levels of semantic relatedness (***Figure 1E***).

Overall, we predicted that relatedness would accelerate learning, benefit later memory, and increase interdependence of linked pairs across lists. Our first analyses focused on overall memory performance, based on proportion of words recalled, across each of five conditions. Following this, we asked whether this changed as a function of semantic relatedness along the respective dimension of each condition (e.g. B-D similarity in the ΔTarget condition) and found the relative memorability of each base pair across subjects by comparing memory for that condition against the Control condition. This allowed us to eliminate the incidental memorability of each individual base pair and focus on how each condition and level of relatedness affected memory. We later visualized these effects as Osgood's proposed surface by plotting the No Δ condition at the cue and target identity point, the ΔTarget condition as a line of values at cue identity spanning target relatedness, the ΔCue condition as a line of values at target identity spanning cue relatedness, and the ΔBoth condition as a surface spanning bivariate cue and target relatedness (see ***Figure 1A*** for the full Osgood surface; see ***Figure 1B*** for a visualization from the narrower stimulus set and ***Figure 1—figure supplement 1*** from the wider set). Later, we examined the speed of learning (based on the number of rounds to reach the criterion of one correct recall per trial) across conditions and levels of relatedness. Finally, we conducted a set of analyses comparing our proactive results from this paper to the retroactive results in our prior paper (***Antony et al., 2022b***). While these results diverge from convention in that they compare groups that were not randomly assigned, they do involve the same stimuli, experimental procedure, timing, and subject population, and we use our control conditions to further adjust for any meaningful differences between the studies (other than learning order). We believe that these similarities make these comparisons suggestive of important differences between proactive and retroactive effects, and we include appropriate caveats to avoid exceptionally strong conclusions. Additionally, to assess whether all retroactive versus proactive differences could be attributed to the fact that we measured memory for the second-learned (base pair) list first at the final test, we ran a control experiment in which base pairs were learned first and supplemental pairs second (i.e. the design from our prior paper) while

testing the supplemental pairs first. We then compared this condition to the same learning condition from our prior paper that tested the lists in the original order.

## Results

### High semantic relatedness resulted in proactive facilitation at a longer delay

Our first analyses focused on how the supplemental pair learning condition affected memory for the later-learned base pairs. In the first two experiments we will discuss, we selected a narrow range of semantic relatedness values using AS from free association norms, or the likelihood that a given word elicits a second word based on a large corpus (*Nelson et al., 2004*). We found supplemental pairs that predicted words from their respective base pairs using AS values of 0.03 (e.g. pious → holy) to 0.96 (e.g. moo → cow). We then tested memory either 5 min after or 48 hr after base pair learning.

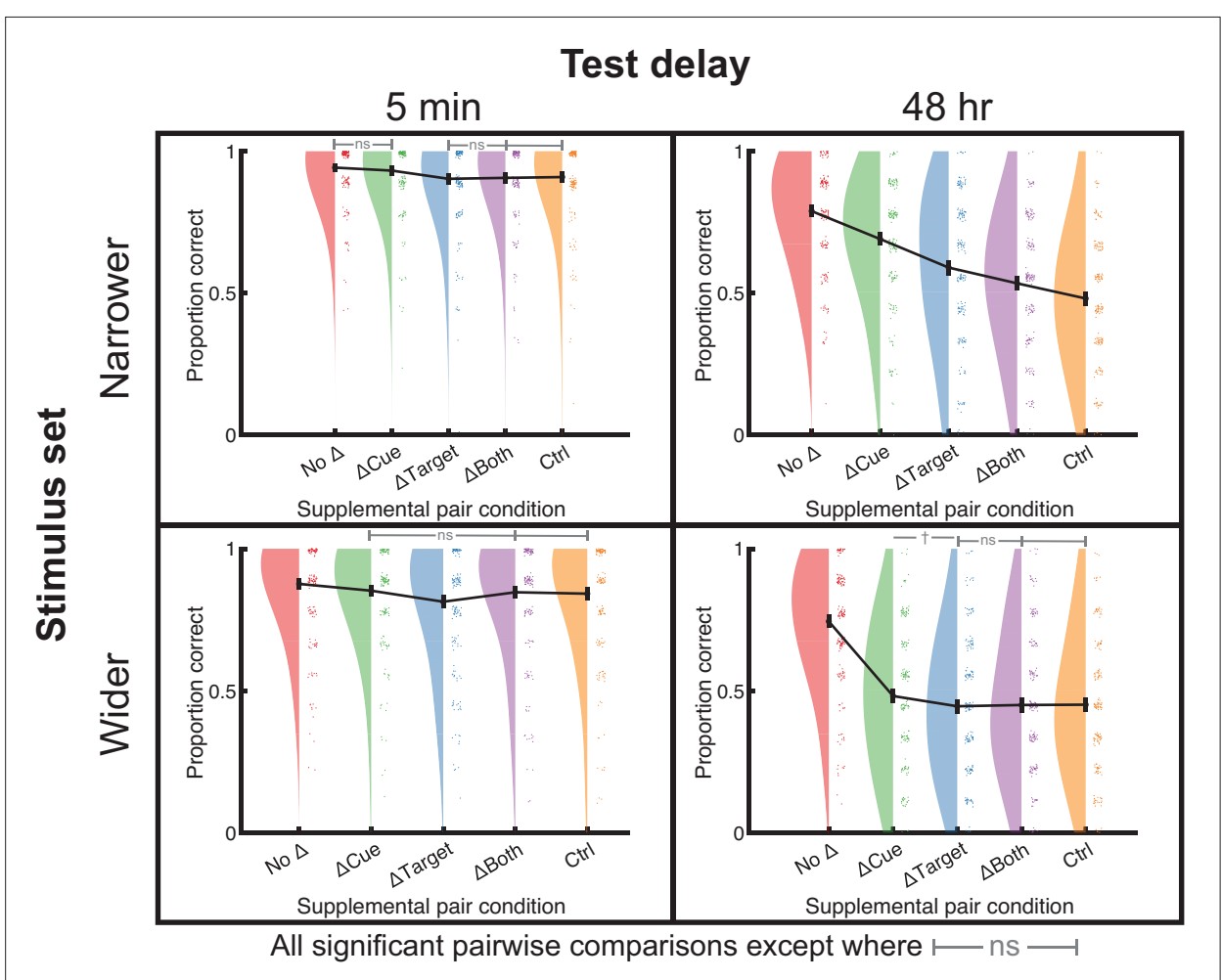

**Figure 2.** Stimulus set, test delay, and experimental condition all determined whether PF or PI occurred. The narrower stimulus set included only single-step semantic relations between base and supplemental cues and targets (top row), whereas the wider stimulus set included the full range of semantic relationships found in the English language (bottom row). Delays were either 5 min (left column) or 48 hr (right column). All comparisons were significant at the $p < 0.05$ via follow-up $t$-tests except those labeled with connected gray bars and either 'ns' ($p > 0.1$) or † ($0.05 < p < 0.1$), and sets of connected bars indicated all 'ns' differences. Data points from individual subjects ($N = 200$ in each experiment) were jittered for improved visualization. See *Figure 2—figure supplement 1* for supplemental pair memory.

The online version of this article includes the following figure supplement(s) for figure 2:

**Figure supplement 1.** Supplemental pair memory differed as a function of overall stimulus set relatedness, delay, and word pair condition.

We measured base pair memory performance across five conditions using a one-way (No Δ, ΔTarget, ΔCue, ΔBoth, and Control) ANOVA.

Focusing first on the 5 min delay condition, we found a significant difference in performance across some conditions [$F(4,796)=7.0$, $p<0.001$] (*Figure 2*, top left) but no evidence of PI relative to the Control condition. Follow-up *t*-test patterns followed a No Δ = ΔCue > ΔTarget = ΔBoth = Control pattern (all significant differences indicate adjusted $p<0.01$; '=' indicates $p>0.27$). In the 48 hr delay experiment, condition again significantly affected overall memory [$F(4,796)=116.3$, $p<0.001$], and follow-up *t*-tests showed significant differences across all pairwise conditions, such that No Δ > ΔCue > ΔTarget > ΔBoth > Control (all adjusted $p<0.003$; *Figure 2*, top right). The finding of higher recall in the conditions with changes relative to the Control condition suggests that the prior learning of semantically related words resulted in PF and that changing the cue benefitted memory more than changing the target. Interestingly, changing both cues and targets resulted in PF, even though there were no words in common across supplemental and base pairs. In tandem, these experiments showed that high semantic relatedness between earlier and more recently learned information created PF at longer delays. They also strongly resemble the retroactive patterns from our prior investigation (*Antony et al., 2022b*) and generally support predictions from Osgood's proposed surface (*Osgood, 1949*).

## Condition and delay altered proactive effects when using a wider range of semantic relatedness

The above results showed either robust PF or no effect in the experimental conditions. Notably, although the ΔTarget condition often generates PI (*Anderson and Neely, 1996*), we found none. Importantly, in these experiments, though cue-Δcue and target-Δtarget associations varied, they were strongly related overall. Thus, these results could have arisen because even the weakest association (e.g. pious → holy) had an AS of 0.03 and was still more related than the average word pair plucked from the overall semantic space of words. In the next two experiments, we therefore expanded semantic relatedness values to pairs spanning the full range of relatedness in the English language. To assess relatedness in this (wider) stimulus set, we found cosine similarity [$\cos(\theta)$] between GloVe vector embeddings (*Pennington et al., 2014*), which were trained on word-word co-occurrences in a large collection of texts and correspond with human judgments of similarity (*Pennington et al., 2014*). We found GloVe values distributed from –0.14 to 0.95, with pairs ranging from unrelated (e.g. sap → laugh) to strongly related (e.g. blue → red). Please note that GloVe is an entirely different relatedness metric and is not a linear transformation of AS (see *Figure 1—figure supplement 1* for how the two stimulus sets compare using the common GloVe metric).

We tested memory for these stimuli using 5 min and 48 hr delays before the final test (*Figure 2*, bottom). In the 5 min delay experiment, the pattern changed. Base pair memory significantly differed across conditions [$F(4,796)=8.50$, $p<0.001$], and there was PI in the ΔTarget condition, such that No Δ > ΔCue = Control = ΔBoth > ΔTarget (ΔCue vs. Control, adjusted $p=0.41$; Control vs ΔBoth, $p=0.63$; all others, $p<0.05$; *Figure 2*, lower left). However, in the 48 hr delay experiment, base pair memory differed strongly across conditions [$F(4,796)=125.2$, $p<0.001$], with pairwise *t*-tests indicating that No Δ > ΔCue >= ΔTarget = ΔBoth = Control (all significant adjusted $p<0.001$; '>=' indicates $0.1>p>0.05$ for ΔCue against all three later conditions; '=' indicates $p>0.8$). Therefore, we found PI effects in the ΔTarget condition, but this was only the case with lower average relatedness and a short retention interval. Please see *Figure 2—figure supplement 1* for results from supplemental pair testing in all experiments.

## Target relatedness produced proactive facilitation and scaffolded new target learning

The preceding effects are partially consistent with Osgood's predictions (*Osgood, 1949*) and our prior paper on retroactive effects using the same stimuli (*Antony et al., 2022b*). That is, under conditions of strong overall relatedness, we found long-term PF in each experimental condition, with greater facilitation in the ΔCue > ΔTarget > ΔBoth conditions, whereas shorter delays and lower overall relatedness levels resulted in the facilitation being reduced or eliminated (although note that ceiling effects were a concern with shorter delays). Moreover, we found classic PI in the ΔTarget condition when there was both a short delay and lower overall relatedness. This result strikingly resembles the pattern

in our prior paper, in which the same combination of experimental condition, timing, and stimulus set was the only one to produce classic RI effects (*Antony et al., 2022b*).

Yet, a more detailed way to investigate Osgood's predictions involves measuring PI vs. PF as a function of semantic relatedness. We began these analyses by looking at memorability in the ΔTarget condition. To calculate memorability, we subtracted the proportion of subjects recalling each word pair in the ΔTarget condition minus the Control condition. This produced a memorability value for each pair that indicates PF when positive and PI when negative. We then regressed memorability against the semantic relatedness of the pair, using either AS or GloVe values.

Interestingly, semantic relatedness between targets produced greater memorability only in the wider stimulus set, 48 hr experiment [$r^2(44)=0.36$, p<0.001; other experiments had non-significant relationships, all $r^2(44)<0.02$, p>0.20] (*Figure 3A*). A careful examination of the regression line also shows that, whereas the ΔTarget condition showed no overall PI or PF relative to the Control condition, we found evidence for *both* PI and PF at different levels of semantic relatedness. For pairs with low relatedness, we found a significantly negative *y*-intercept (p<0.001), indicating PI for these pairs, but for pairs with high relatedness, this trend reversed to PF. This was verified by splitting pair relatedness values into thirds and running one-sample *t*-tests for all words in each third against 0. The lowest third showed PI [$t(14)=-3.6$, p=0.003], whereas the upper third showed PF [$t(14)=4.4$, p<0.001].

The lack of effect in the 5 min experiments could be due to ceiling effects, as performance was high in both experiments. However, the lack of effect in the 48 hr, narrow stimulus set was surprising, as the retroactive equivalent was significant in our prior study (*Antony et al., 2022b*). We considered that one important difference between the studies was the directional nature of the AS values (*Sahakyan and Goodmon, 2007*). In our prior study, we reasoned that we should use AS from supplemental → base pairs, such that the supplemental pairs would retroactively act upon the main memory of interest. However, given that the benefits could arise due to remindings, one could reason that this directional process is more important for a later-learned → earlier-learned representation than the converse. For this reason, we also investigated the relationship using base → supplemental pair AS. This relationship was also not significant [$r^2(44)=0.007$, p=0.41]. We will return to this surprising result in the Discussion.

Under predictions from a remindings-based account of proactive effects of memory (*Wahlheim and Jacoby, 2013*), recalling a prior pair during new learning can boost memory for pairs by creating interdependence among elements from earlier and more recent pairs. We next investigated the interdependence between corresponding initial (cue-target) and later-learned (cue-Δtarget) pairs. We operationalized interdependence as the frequency by which base pair target-supplemental pair target duos across subjects were either both correct or both incorrect across subjects (*Horner et al., 2015*). That is, interdependence means that recalling the cue-target pair increases the likelihood of recalling the cue-Δtarget pair (relative to a threshold of matching up one pair against all other pairs). Next, we correlated interdependence against relatedness. We found that interdependence significantly increased with relatedness in the 48 hr experiments [both, $r^2(44)>0.11$, p<0.02], and this was marginally significant in the 5 min experiments [both $r^2(44)<0.06$, 0.07<p < 0.08] (*Figure 3B*). Overall, our investigations of longer delays under a wider range in relatedness values showed that target relatedness correlated with base pair memorability and interdependence between base and supplemental pairs. Under other conditions, the relationship was weaker or non-significant, although the relationship between relatedness and interdependence was more reliable than memorability.

Additionally, we investigated whether target relatedness increased intrusions, operationalized here as entering the supplemental pair list response during the base pair test. Importantly, this helped us contrast between predictions from a PI-based account of our results and predictions from a successful remindings account of proactive memory effects. Under the PI-based account, targets could merge into a unified representation that does not allow the subject to differentiate the contexts (e.g. peaceshaverazor [written jointly to emphasize the unification]; *McCrystal, 1970*; *Postman and Keppel, 1977*), as some studies have shown greater intrusion errors with increasing relatedness among stimuli (*Dallett, 1962*; *Dallett, 1964*; *Osgood, 1946*; *Underwood, 1951*). This account is not necessarily contradicted by the preceding results, as the merged representation could (on average) help maintain the long-term memory while also leading to occasional source confusions. Conversely, the remindings-based account predicts that successful retrieval of existing memories during new learning scaffolds old and new information (leading to interdependence) while allowing subjects to remember the different contextual associations. We adjudicated between these accounts by correlating across-subject

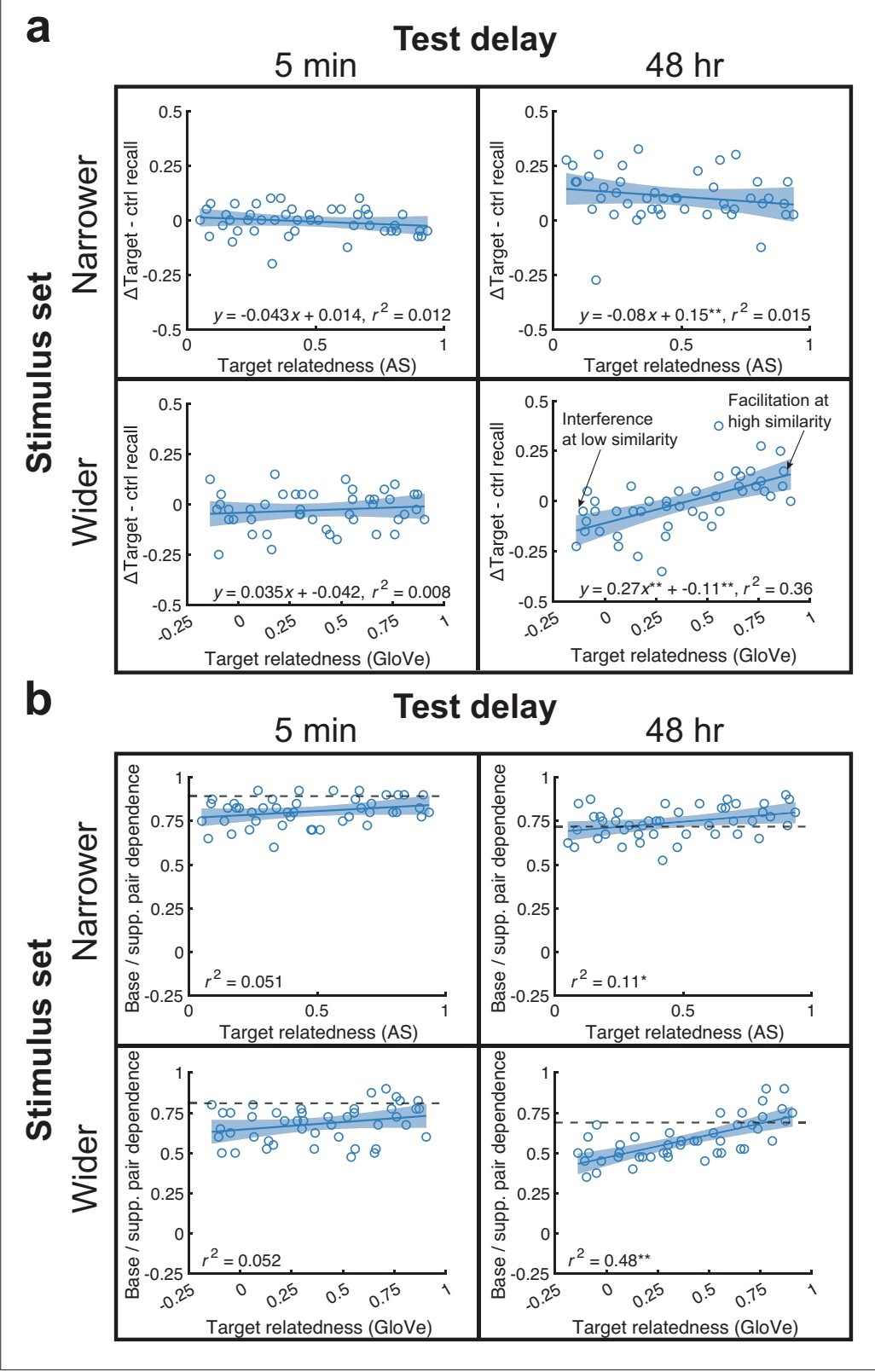

**Figure 3.** Memory and interdependence in the ΔTarget condition as a function of relatedness. (**a**) We plotted each pair's across-subject memorability value against its corresponding target-Δtarget semantic relatedness in all experiments. AS and GloVe values were used in the top and bottom rows, respectively. In the wider relatedness stimulus set, 48 hr experiment, semantic relatedness proactively improved memory. Furthermore, PI occurred

*Figure 3 continued on next page*

*Figure 3 continued*

under conditions of low relatedness and PF under high relatedness. (**b**) We plotted memory dependence for each base pair-supplemental pair target duo against its relatedness in all experiments. High dependence means that subjects tended to remember or forget both targets in the duo (relative to recalling one and not the other). Relatedness significantly increased dependence in both 48 hr experiments ($p < 0.05$) but this was marginally significant in the 5-min experiments ($0.05 < p < 0.1$). Thick dotted lines show the threshold of dependence levels against all other pairs at the 95th percentile. All analyses were based on correlations across 45 base pair memorability or interdependence values and used a significance level of $p < 0.05$. See also *Figure 3—figure supplement 1* for intrusion data from this condition.

The online version of this article includes the following figure supplement(s) for figure 3:

**Figure supplement 1.** Intrusions did not increase with target relatedness in the ΔTarget condition in any experiment.

---

supplemental-into-base pair intrusions against relatedness. Critically, in the wider stimulus set, 48 hr experiment, intrusions significantly decreased with relatedness [$r^2(44)=0.43$, p<0.001], whereas they did not significantly increase in any experiment [all others, $r^2(44)<0.02$, p>0.26]. This finding, in combination with highly similar non-significant increases or significant decreases in intrusions in our retroactive study (*Antony et al., 2022b*), supports predictions from the remindings account (*Wahlheim et al., 2021*) that remindings allow one to build interdependent, yet contextually discriminable representations.

## Cue relatedness significantly increased base pair memorability and interdependence

In the above analyses comparing memory in the ΔCue versus Control conditions, we found PF in all experiments. Next, we asked whether the extent of these PF effects increased with increasing relatedness among cues linked with the same target. We found that relatedness increased memorability in both 48 hr experiments [both $r^2(44)>0.09$, p<0.02] but neither 5 min experiment [both $r^2(44)<0.02$, p>0.44] (*Figure 4A*). Interestingly, in the wider stimulus set, 48 hr experiment, we found a similar effect as in the ΔTarget condition analyses, in that there was PI at low relatedness and PF at high relatedness. Splitting relatedness values into thirds, the lowest third demonstrated significant PI [$t(14)=3.0$, p=0.009], whereas the upper third demonstrated PF [$t(14)=3.5$, p=0.003]. These results differ somewhat from our prior paper (*Antony et al., 2022b*), which showed similar facilitation in the ΔCue condition, but in which most correlations between memorability and relatedness were positive yet fell short of significance. Here, these relationships were significant in the 48 hr experiments. Another difference is that, whereas we did not find any reliable interference in the ΔCue condition in the prior study, there was significant PI in the low relatedness items in the wider stimulus set, 48 hr experiment. The *y*-intercept in the identical condition of the prior experiment was significantly above, rather than below, zero.

Additionally, we measured interdependence between target memory in the base and supplemental pair conditions. Interdependence increased in all experiments [all $r^2(44)>0.08$, p<0.04].

## Cue + target relatedness increased memory accuracy and interdependence in the ΔBoth condition

In the preceding analyses, we found PF in the ΔBoth condition in the narrower stimulus set, 48 hr delay experiment but not in the other experiments. Next, we investigated whether memorability and/or dependence increased as a function of cue +target relatedness. Unlike in our prior retroactive experiment, we found no effects of relatedness on memorability [all $r^2(44)<0.03$, p>0.48], even though overall memory was better in the ΔBoth condition than the Control condition in the narrower stimulus set, 48 hr delay experiment. Conversely, we found that relatedness improved interdependence in all experiments [all $r^2(44)>0.07$, p<0.04] except the wider stimulus set, 5 min delay experiment [$r^2(44)=0.02$, p>0.85]. Therefore, although there was no evidence that relatedness strengthened base pair memory, it generally promoted interdependence between base and supplemental pair memory. Overall, our results somewhat resemble the retroactive effects in *Antony et al., 2022b* in that both papers showed significant overall facilitative effects and correlations between cue +target relatedness interdependence in the 48 hr, narrower stimulus set experiment. However, in the prior study, correlations between cue +target relatedness with memorability were also significant. Note that our

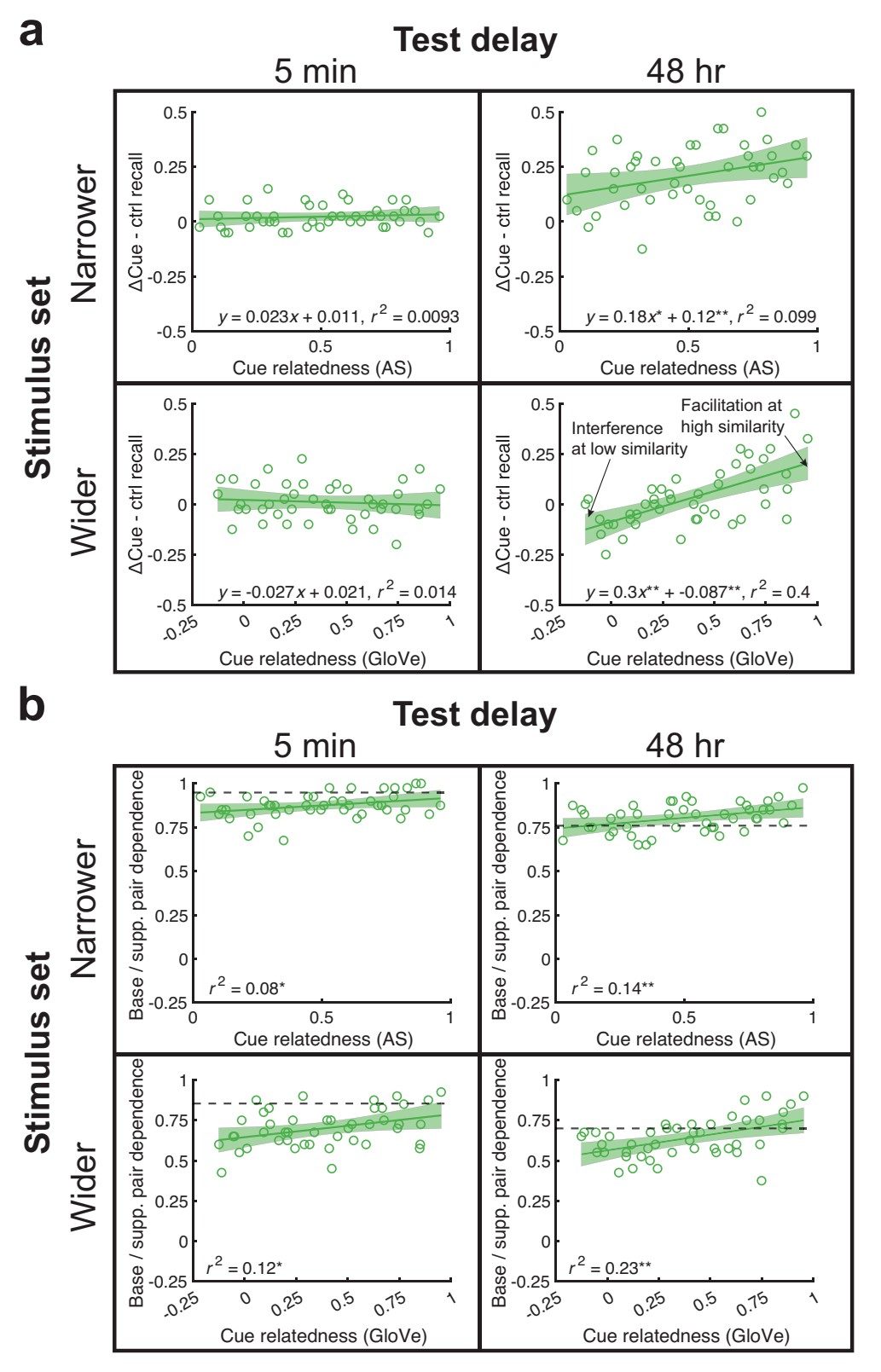

**Figure 4.** Semantic relatedness between cues benefits memory in the 48 hr experiments and boosts interdependence in all experiments. (**a**) Across-subject memorability for each pair in the ΔCue minus Control condition benefited from cue relatedness in the 48 hr, but not 5 min, experiments (top: AS; bottom: GloVe). (**b**) Relatedness increased memory interdependence in all experiments. Thick dotted lines show the threshold of

*Figure 4 continued*

dependence levels against all other pairs at the 95th percentile. All analyses were based on correlations across 45 base pair memorability or interdependence values and used a significance level of $p < 0.05$.

prior paper featured a smoothed, 2-D plot with cue and target relatedness as separate predictors and memorability or dependence as the outcome variable (the cue +target relatedness plot was significant but was provided as a supplement). We have provided the smoothed, 2-D plot as *Figure 5*, *Figure 5—figure supplement 1* and as part of the next subsection of the Results.

## Osgood-style surfaces for proactive memorability and interdependence

One motivating factor for designing this study with such large sample sizes, five conditions, and continuous relatedness ratings was to re-create Osgood's predicted surface in one consolidated figure. We therefore re-plot results from all experimental conditions against the Control condition in *Figure 6*. In these plots, the strength of any pair lies along the *x-y* plane, which reflects the target and cue relatedness of its corresponding supplemental pair, respectively. If there was no preceding supplemental pair (Control condition), the pair remains on the *x-y* plane (orange). If the base pair is a repeat of the supplemental pair (No Δ condition), it belongs to the target identity, cue identity point (red). If only targets change (ΔTarget condition), the pair lies along the cue identity line according to its target relatedness (blue), whereas if only cues change (ΔCue condition), the pair lies along the target identity line according to its cue relatedness. Lastly, if both cues and targets change (ΔBoth condition), the pair belongs on a 2-D surface as a function of both cue and target relatedness (purple).

Using similar formatting, we also plotted memory interdependence between base and supplemental pairs. Generally, relatedness increased with dependence along multiple dimensions in a stronger manner than its effects on proactive memory strengthening. These surfaces attempt to characterize a large range of factors that may proactively impact later memory formation and determine when earlier and more recent memories become linked.

## Re-assessing proactive memory and interdependence effects using a common metric

The above stimulus sets intentionally spanned different ranges of relatedness, but we wanted to ensure our effects held when using a common metric. Therefore, we combined across-subject memorability and interdependence values across stimulus sets in each experimental condition, keeping the results separate for a given test delay. In other words, we combined the two 5 min delay experiments together and the 48 hr delay experiments together. Then, we correlated memorability with GloVe relatedness values in a similar fashion to the above analyses (*Figure 6—figure supplement 1*). In the 5 min delay experiment, there were no significant correlations between memorability and relatedness in the three conditions [all $r^2(89)<0.011$, p>0.2]. Correlations between interdependence and relatedness were significant in the ΔCue [$r^2(89)=0.19$, p<0.001] condition and ΔTarget condition [$r^2(89)=0.14$, p<0.001], but not the ΔBoth condition [$r^2(89)=0.02$, p=0.12]. In the 48 hr delay experiment, correlations between memorability and cue relatedness in the ΔCue condition [$r^2(89)>0.29$, p<0.001] and target relatedness in the ΔTarget condition [$r^2(89)=0.2$, p<0.001] were significant, whereas cue +target relatedness in the ΔBoth condition was not [$r^2(89)=0.01$, p=0.58]. In all three conditions, interdependence increased with relatedness [all $r^2(89)>0.16$, p<0.001].

## Semantic relatedness accelerated new learning

Various paradigms have shown that new learning is accelerated when it can be linked with prior information (*Barnes and Underwood, 1959*; *Bartlett, 1932*; *Bein et al., 2015*; *Brod et al., 2013*; *Jarrett and Scheibe, 1963*; *Metcalfe et al., 1993*; *Morton et al., 2023*; *Palermo and Jenkins, 1964*; *Postman and Parker, 1970*; *Tse et al., 2007*; *Underwood, 1951*; *van Kesteren et al., 2012*; *Wimer, 1964*; *Witherby and Carpenter, 2022*; *Young, 1955*). Indeed, we found this to be the case in our prior experiment, such that relatedness within each condition predicted fewer rounds to achieve learning criterion for later-learned pairs (in that case, supplemental pairs, which were learned after base pairs) (*Antony et al., 2022b*). Here, we had the opportunity to perform the same analyses but also compare results to a later-learned Control condition. Using average number of trials to criterion during base

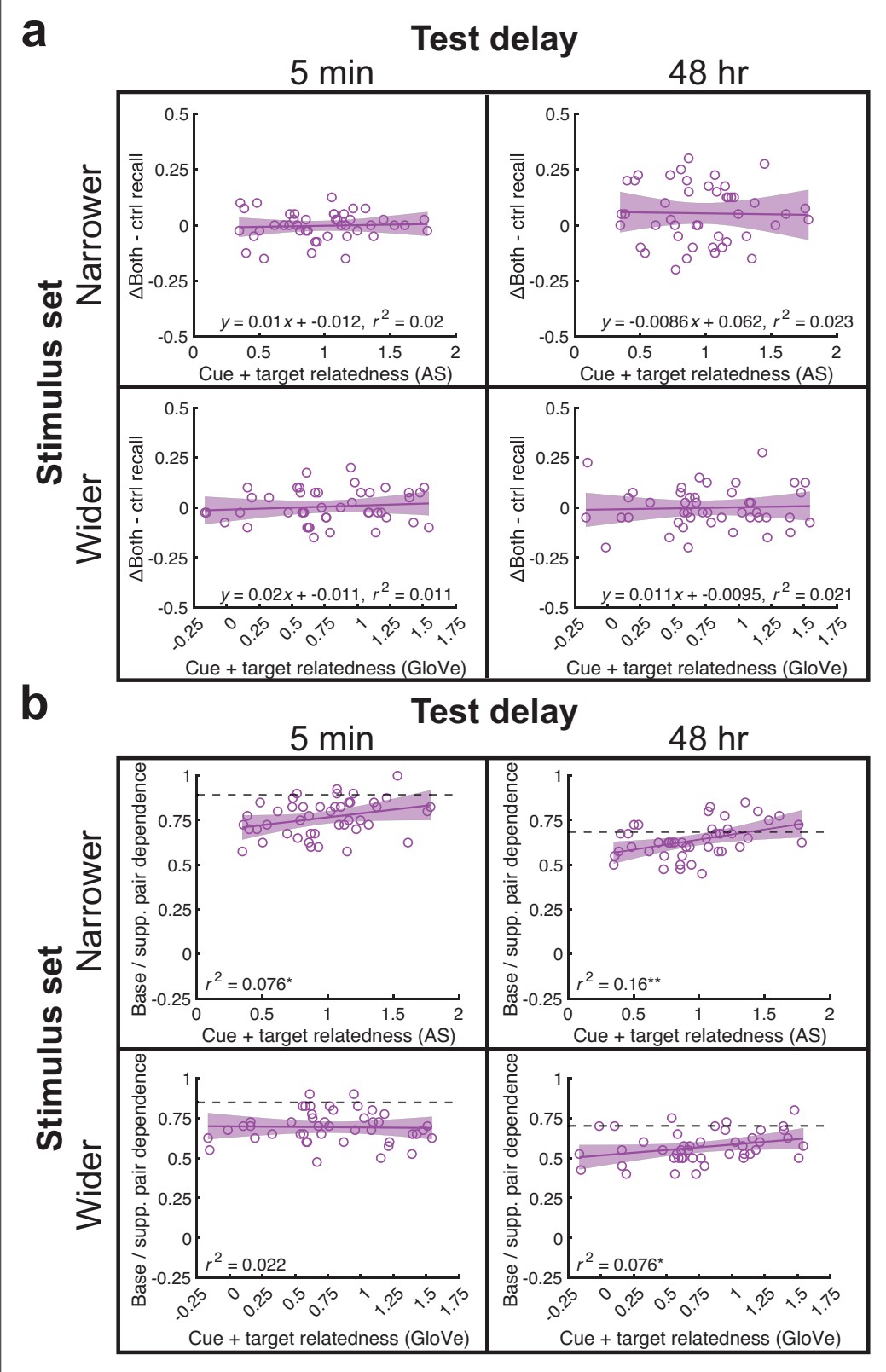

**Figure 5.** Increasing cue +target relatedness boosted interdependence but not memorability in the ΔBoth condition across experiments. (**a**) We plotted cue +target relatedness against memorability in each experiment. Memorability did not increase with relatedness in any experiment. (**b**) Memory interdependence increased with cue +target relatedness in all experiments except the wider stimulus set, 5 min delay experiment. All analyses were

*Figure 5 continued on next page*

*Figure 5 continued*

based on correlations across 45 base pair memorability or interdependence values and used a significance level of *p* < 0.05. See *Figure 5—figure supplement 1* for smoothed, 2-D plots that plot cue and target relatedness as separate variables predicting memorability and interdependence.

The online version of this article includes the following figure supplement(s) for figure 5:

**Figure supplement 1.** Bivariate cue and target relatedness had no effect on memorability, though it did significantly predict interdependence in the narrower stimulus set in the 48 hr delay experiment.

pair learning as an outcome measurement, we evaluated overall condition effects as well as correlations with relatedness. In the narrower stimulus set experiments, the number of trials to criterion followed a No Δ < ΔCue < ΔTarget < ΔBoth < Control condition pattern (5 min delay: all comparisons, adjusted p<0.008; 48 hr delay: all comparisons, adjusted p<0.007; *Figure 7A*). The wider stimulus set experiments produced a somewhat similar pattern of No Δ < ΔCue <= ΔTarget < ΔBoth = Control condition pattern (5 min delay: ΔBoth vs. Control, adjusted p=0.92; all other comparisons, adjusted p<0.03; 48 hr delay: all comparisons except ΔBoth vs. Control and ΔCue vs. ΔTarget, adjusted p<0.01; ΔBoth vs. Control, p=0.19; ΔCue vs. ΔTarget, p=0.19). These results resemble the overall memory results in a condition-by-condition fashion, such that faster learning within a condition generally led to better memory after a 48 hr delay. Next, we correlated the average base pair trials to criterion across subjects as a function of relatedness in the ΔCue, ΔTarget, and ΔBoth conditions. We found that higher cue relatedness produced faster base pair learning in every experiment, either significantly or marginally significantly [narrower stimulus set, 48 hr delay experiment: $r^2$(44)=0.064, p=0.052; all others, $r^2$(44)>0.065, p<0.05; (*Figure 7B*)]. Similarly, higher target relatedness produced faster base pair learning in every experiment, either significantly or marginally significantly [narrower stimulus set, 5 min experiment: $r^2$(44)=0.04, p=0.09; all others, $r^2$(44) = >0.11, p<0.02] (*Figure 7C*). Finally, additive cue +target relatedness generally produced faster base pair learning in the narrower stimulus set experiments [5 min delay experiment: $r^2$(44)=0.08, p=0.04; 48 hr delay experiment: $r^2$(44)=0.046, p=0.08] but not the wider stimulus set experiments [5 min delay experiment: $r^2$(44)=0.0045, p=0.37; 48 hr delay experiment: $r^2$(44)=0.024, p=0.16; (*Figure 7D*)]. Overall, it appears that previously learned supplemental pairs scaffold and hasten learning of base pairs depending on their level of relatedness, with most relationships either significant or marginally significant.

## Comparing retroactive and proactive memory effects

The preceding results accord well with the qualitative results from our prior study examining retroactive effects (*Antony et al., 2022b*; with a few exceptions, which we cover more extensively in the Discussion). One conclusion from these converging results could be that relatedness benefits memories in the retroactive and proactive case for similar reasons, namely that by virtue of increasing the likelihood of looking back and associating new learning with existing memories, it helps the connected memories survive due to being embedded in an integrated memory structure (*Antony et al., 2024*; *Antony and Schechtman, 2023*; *Lee and Chen, 2022*; *Schlichting et al., 2015*; *Trabasso and Sperry, 1985*). Our results here and earlier, as well as those from other recent studies showing that memory updating benefits from variables that improve contact between the past and present (e.g. *Kemp et al., 2023*; *Wahlheim et al., 2020*; *Wahlheim et al., 2023*; *Wahlheim and Zacks, 2019*), support this idea.

Nonetheless, one could still conclude that there should be asymmetric benefits: remindings are retrieval events, and retrieval practice profoundly affects memory (*Antony et al., 2017*; *Gates, 1917*; *Roediger and Karpicke, 2006*; *Rowland, 2014*). Therefore, existing memories may receive the additional benefits of an extra retrieval trial, while new learning does not receive that benefit. In this final section, we will adjudicate between these predictions by comparing overall base pair memory from the proactive conditions here against the comparable retroactive conditions from *Antony et al., 2022b*. We highlight the caveat that we conducted these experiments at different times. Subjects were therefore not randomly assigned to proactive and retroactive effects conditions. Nevertheless, both studies used the same subject population, learning materials, learning criteria, and timing across experiments, and we will additionally use internal control measurements to make these comparisons as equal as possible.

To compare proactive and retroactive effects directly, we sought a better measurement than raw memory performance in each condition that would allow us to account for within-participant

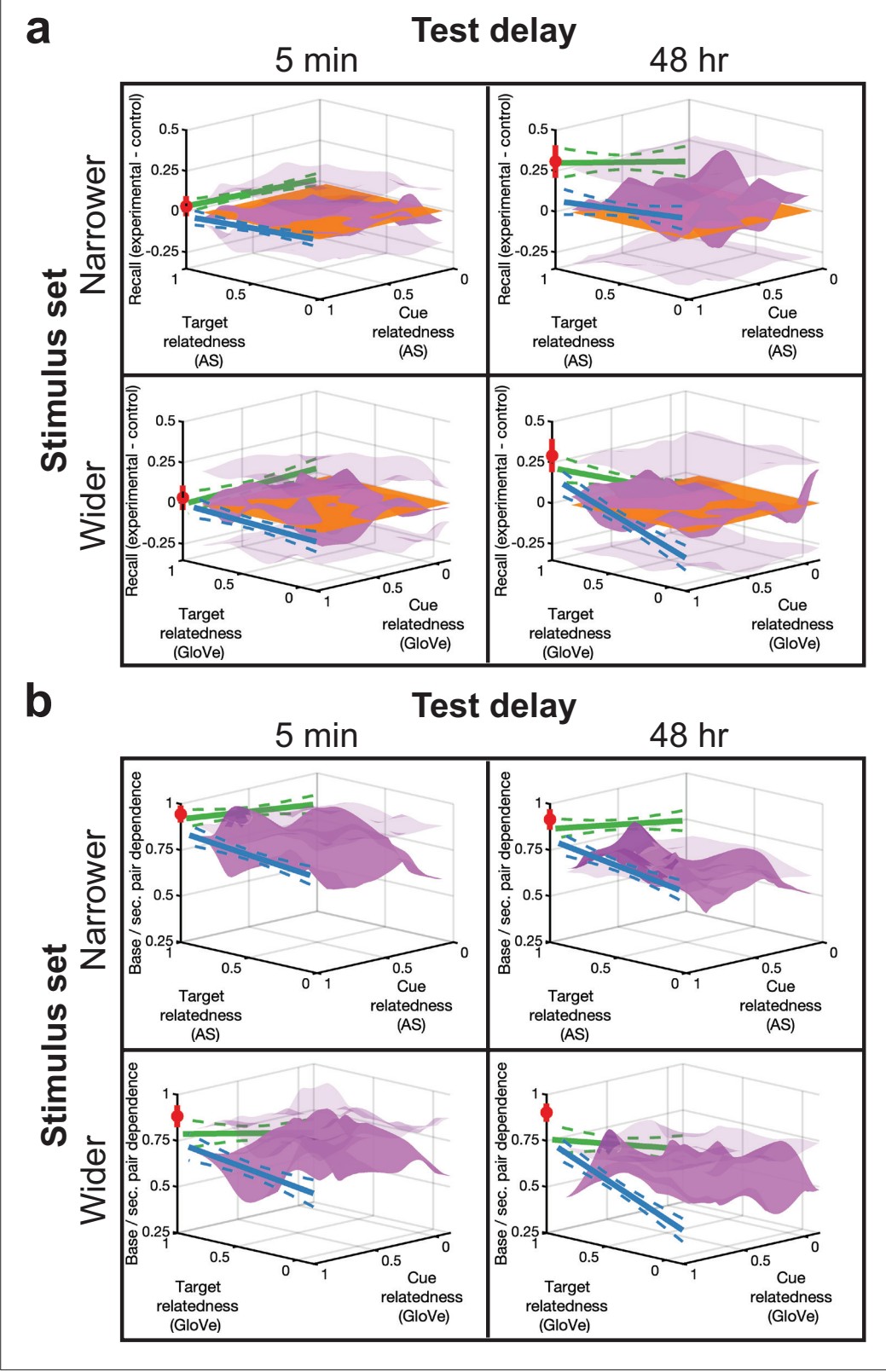

**Figure 6.** Osgood-style surfaces for proactive effects and interdependence. (**a**) We plotted memorability from all conditions minus the Control condition from all experiments in 3-D coordinates. These plots had cue and target relatedness on the *y*- and *x*-axes, respectively. On the *z*-axis, we plotted base pair memorability, oriented as PF for positive change and PI for negative change. Memorability for the No Δ (±across pair standard deviation) is

*Figure 6 continued on next page*

*Figure 6 continued*

at the cue identity, target identity corner point (red circle). We plotted ΔTarget condition memorability along the cue identity line against target relatedness (blue) and ΔCue condition memorability along the target identity line against cue relatedness (green) (both using ± standard error from the ordinary-least-squares regression fit). We plotted ΔBoth condition memorability as a locally smoothed surface as a function of both cue and target relatedness (purple). Transparent surface grids above and below zero represent p<0.01 significance boundaries from permutation tests; significant points on the surface are indicated by a darker shade of purple. (**b**) Interdependence measurements for all experiments and conditions are formatted similarly to (**a**).

The online version of this article includes the following figure supplement(s) for figure 6:

**Figure supplement 1.** Memory and interdependence data from preceding analyses plotted using a common GloVe metric.

performance. As opposed to results in the preceding sections, we believe here that the best control is the positive control (No Δ condition), because, given that list-based interference has been reported in some studies using A-B, C-D, test C-D designs (*Postman and Keppel, 1977*), it is possible that list-based interference still affected new learning in the (negative) Control condition to a small degree. Therefore, we examined memory in all conditions minus the (always superior) No Δ condition, wherein less negative values indicated better memory. We also eschewed the 5 min delay experiments, because in those cases, there is a considerable difference between the timing since base list learning. That is, it was only 5 min in the proactive case, whereas it was 5 min plus the time it took to learn supplemental pairs in the retroactive case; given this difference, one would expect memory differences. However, we include those comparisons in *Figure 8—figure supplement 2* for those interested.

In comparing retroactive and proactive designs in the narrower stimulus set, 48 hr experiments, we found better memory in the retroactive experiment in the ΔCue, ΔTarget, and ΔBoth conditions (all adjusted p<0.011) and not the (negative) Control condition (p=0.43; *Figure 8*, top left). In the wider stimulus set, 48 hr experiments, there were retroactive advantages in the ΔCue and ΔTarget conditions (both p<0.005) but neither the ΔBoth (p=0.31) nor the Control conditions (p=0.94). Therefore, all conditions showing long-term memory facilitation also showed greater facilitation in the retroactive than proactive case (ΔCue, ΔTarget, and ΔBoth in the narrower stimulus set; ΔCue and ΔTarget in the wider stimulus set; *Figure 8*, top right).

Finally, we wanted to run one more control experiment to rule out a potential difference between the retroactive and proactive designs that the preceding comparisons could not address. Specifically, one design involves subjects first recalling items from a first list and the other a second list. It remains possible that recall is slightly more difficult when mentally flipping around everything one learned after a certain delay. To address this concern, we ran another experiment (N=200) with the following order: base pair learning, supplemental pair learning, 48 hr delay, supplemental pair testing, base pair testing. We refer to this as the flipped test experiment and it used the narrower stimulus set. In this experiment, we calculated supplemental pair memory in each condition minus the No Δ condition, and we compared it in the flipped test experiment (when tested first after the delay) to the other narrower stimulus set, 48 hr, retroactive experiment (when it was tested second after the delay). Contrary to the idea that probing memory out of order impairs it, we found non-significantly better memory in the flipped test version in the ΔTarget and ΔBoth (both 0.12<p < 0.13) conditions and no differences in the ΔCue condition (p=0.94).

This experiment also served two other purposes. First, because this flipped test experiment was conducted online, it allowed us to control for experimental setting when comparing the narrower stimulus set, 48 hr experiments, because the original retroactive, narrower stimulus set, 48 hr experiment took place in person (before the Covid-19 pandemic). Second, it replicated the main findings in our prior study (see *Figure 8—figure supplement 1* for average results from each condition in this flipped test experiment) and allowed us to replicate retroactive versus proactive comparisons described above. We therefore contrasted base pair memory (subtracting No Δ condition memory) in this flipped test experiment against base pair memory in the proactive experiment. We again found better memory in the retroactive experiment in the ΔCue, ΔTarget, and ΔBoth conditions (all adjusted p<0.002) and not the (negative) Control condition (p=0.52; *Figure 8*, bottom left). Finally, we directly compared base pair memory in the retroactive, narrower stimulus set, 48 hr delay experiment from our prior study against base pair memory in the flipped test experiment. We found no significant

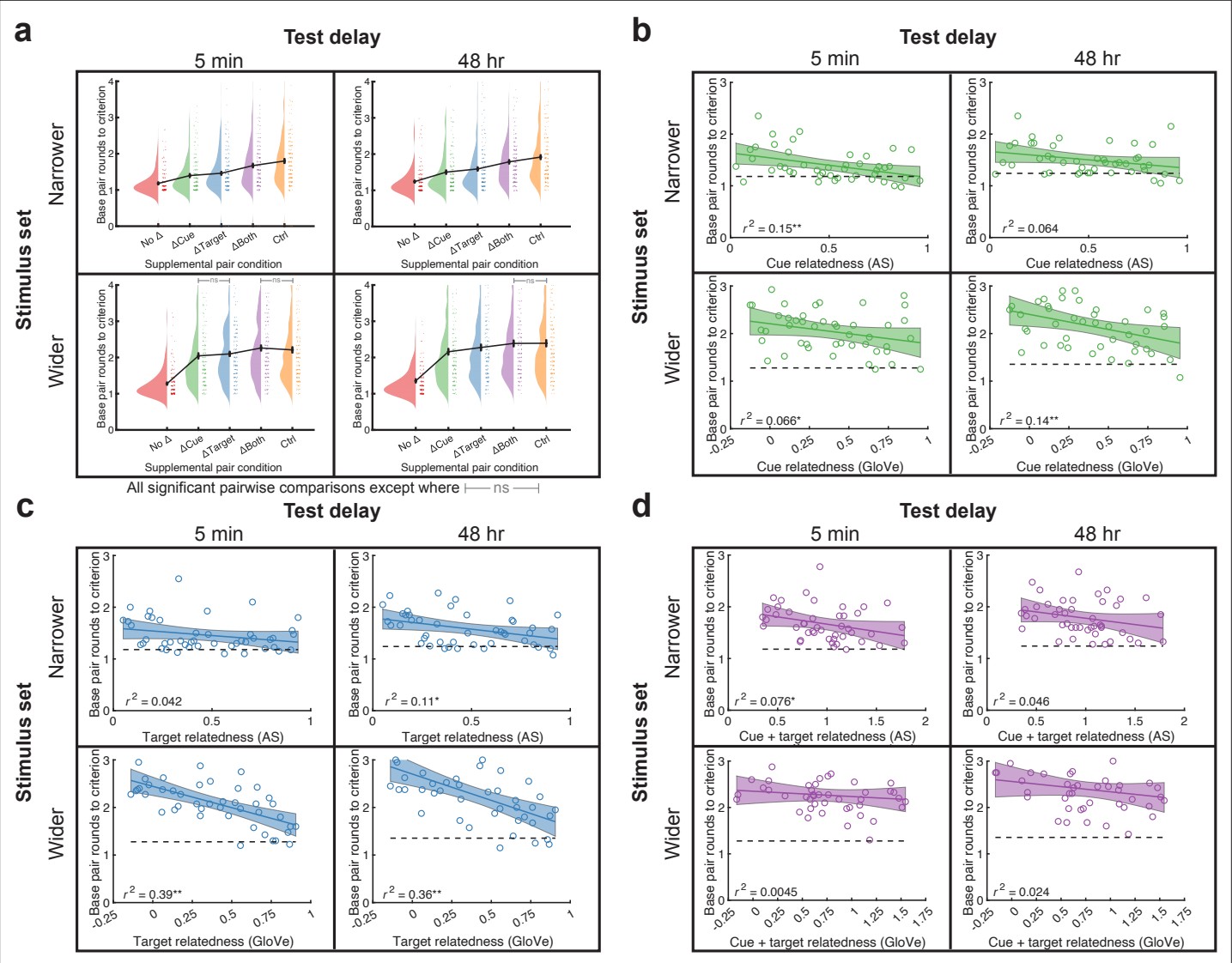

**Figure 7.** Base pair learning differed based on stimulus set and condition and generally benefited from higher semantic relatedness. (**a**) In the narrower stimulus set, learning time (mean trials to criterion) followed this trend: No Δ Both <Control. The wider stimulus set tended to show nearly the same pattern: No Δ < ΔCue </ = ΔTarget < ΔBoth = Control, where '</=' indicates a significant difference in one experiment but not the other. All comparisons were significant except those labeled with gray bars and 'ns' (p>0.1). Data points from individual subjects were jittered slightly for visualization, and a few outliers (slower learners) were omitted from the plot for better visualization. (**b–d**) We correlated relatedness with learning times (top: AS; bottom: GloVe). (**b–c**) Learning time across subjects for each word pair generally decreased with increasing cue relatedness in the ΔCue condition (**b**) and decreased with increasing target relatedness in the ΔTarget condition (**c**). (**d**) In the ΔBoth condition, learning time generally decreased with cue +target relatedness in the narrower stimulus set, but not in the wider stimulus set. In (**b–d**), Pearson correlations are shown in the plots followed by * when p<0.05 and ** when p<0.01. All analyses were based on correlations across 45 base pair rounds to learning criterion values and used a significance level of $p < 0.05$.

differences in any of the ΔCue, ΔTarget, ΔBoth, or Control conditions (all adjusted p>0.87; *Figure 8*, bottom right).

## Discussion

We found that, as old and new information became more semantically related, memory for new information increased and the two grew more interdependent. When considering overall memory across experimental conditions, an overall high level of relatedness between old and new information (as in the narrower stimulus set) resulted in profound long-term PF for the ΔCue, ΔTarget, and ΔBoth

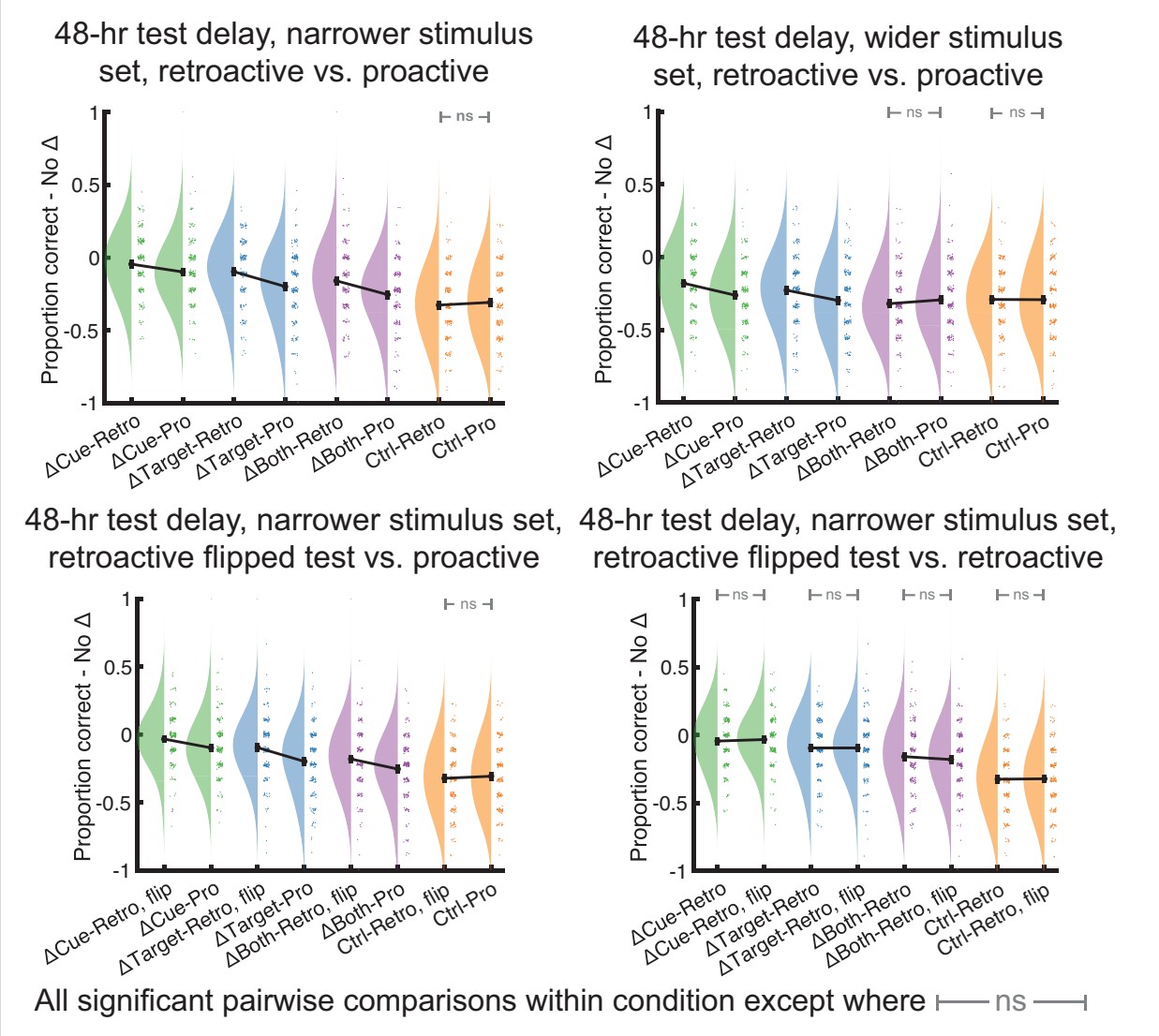

**Figure 8.** Under otherwise identical conditions, memory benefits after 48 hr were stronger for experimental groups under retroactive versus proactive designs when compared to the No Δ condition, which served as a positive control. Comparisons of the same base pair condition across experiments were performed between the narrower stimulus set, retroactive versus proactive experiments (top left); the wider stimulus set, retroactive versus proactive experiments (top right); the narrower stimulus set, flipped test experiment (in which base pairs were subject to retroactive effects) versus proactive experiment (bottom left); and among the narrower stimulus set retroactive experiment versus flipped experiment (bottom right). In all cases, experimental groups that showed facilitation relative to the Control condition in the overall analyses showed greater retroactive than proactive facilitation. All analyses used between-subject *t*-tests and used a significance level of $p < 0.05$.

The online version of this article includes the following figure supplement(s) for figure 8:

**Figure supplement 1.** Memory and interdependence data in a paradigm in which we used base pair learning followed by supplemental pair learning and then flipped the order of the tests (supplemental pairs followed by base pairs).

**Figure supplement 2.** Direct comparisons of the same base pair condition across 5 min delay experiments were performed between the narrower stimulus set, retroactive versus proactive experiments (left) and the wider stimulus set, retroactive versus proactive experiments (right).

conditions relative to the Control condition. When the overall level of relatedness was lower (in the wider stimulus set), we found classic PI effects in the ΔTarget condition in the 5 min delay experiment. Additionally, in the 48 hr delay experiment, the overall memory benefits were smaller (ΔCue condition) or eliminated (ΔTarget condition), but we found an intriguing pattern by which PI occurred with low relatedness and PF with high relatedness in both the ΔCue and ΔTarget conditions. We did not find relatedness correlations with memorability in the ΔBoth condition of any experiment. Additionally,

we found benefits of relatedness on interdependence in all experimental conditions of all 48 hr delay experiments but weaker or less reliable effects in 5 min delay experiments.

These results converged strongly with those from our prior study (*Antony et al., 2022b*) focusing on retroactive memory effects. In both studies, we found overall benefits of each experimental condition in the narrower stimulus set, 48 hr delay experiment. In both studies, we found correlations between relatedness and memorability in the ΔCue and ΔTarget conditions in the wider stimulus set, 48 hr delay experiment, though this was less reliable in the ΔCue condition of the prior study. In fact, in the prior study, you could directly see how delay and relatedness combined to determine retroactive facilitation versus interference. In the ΔTarget condition of the wider stimulus set experiments, retroactive effects went from interference with low relatedness to no interference with high relatedness in the 5 min experiment to no interference with low relatedness to facilitation with high relatedness in the 48 hr experiment. Here, we found this crossover from PI with low relatedness and PF with high relatedness in the same experiment. Therefore, in both studies, the only condition showing overall RI and PI was in the wider stimulus set, 5 min delay condition, supporting the idea that RI and PI are relatively limited to these cases. Thus, the tendency to study memory with short delays and without interrelationships among study material led to the predominance of this idea in interference studies and theories throughout the 20th century. Additionally, in both studies, we also found correlations between relatedness and interdependence in all experimental conditions in the narrower stimulus set, 48 hr delay experiment and the ΔCue and ΔTarget conditions of the wider stimulus set, 48 hr delay experiments. This finding also supports recent studies showing that interdependence increases with contextual similarity between episodes (*Cox et al., 2021*; *Yu et al., 2023*), which may also be more likely to become integrated during encoding.

There were also some differences between the studies. Here, we found an unexpected null relationship between relatedness and memorability in the ΔTarget condition of the 48 hr, narrower stimulus set condition that was significant in the prior experiment. Notably, this was not simply a function of the direction of the associative strength (AS) measure, because flipping around the direction of AS from supplemental target → base target to base target → supplemental target also produced nonsignificant results. We note that there's an overall ΔTarget benefit in this condition, and the *y*-intercept of *Figure 3A*, top right, is significantly above zero, reflecting memorability benefits even with 0 AS. However, the precise level of relatedness was not predictive of memorability benefits. We also found a weaker effect in the ΔBoth condition of the narrower stimulus set, 48 hr delay experiment, such that additive cue +target relatedness did not produce significant memorability benefits (although it still predicted interdependence). Critically, we also found overall differences when directly comparing memory in the otherwise identical retroactive and proactive designs. These comparisons consistently resulted in larger retroactive than proactive memory benefits in each of the experiments showing overall memory facilitation (ΔCue, ΔTarget, and ΔBoth in the narrower stimulus set; ΔCue and ΔTarget in the wider stimulus set). With the caveat that these comparisons do not use random assignment, these highly consistent results support the idea that, while remindings produce benefits for both old and new information, they disproportionally benefit old information, likely due to the additional retrieval trial (*Carpenter et al., 2008*; *Gates, 1917*; *Roediger and Karpicke, 2006*; *Rowland, 2014*).

We will now offer a cumulative explanation of our results, starting with key points about the recursive remindings hypothesis (*Hintzman, 2011*). A series of studies using paired associate learning tasks akin to those here have shown that encouraging subjects to recall existing memories during new learning can reduce PI and promote PF, depending on how well the information is integrated and later recollected (for a review, see *Wahlheim et al., 2021*). In these cases, subjects must both notice and subsequently recollect a change between prior and new information to gain these benefits. Similar findings have been observed when using measures of recollecting change at test to infer noticed changes that were not measured during new learning (*Wahlheim, 2015*). Additionally, remindings promote interdependence between the old and new information while preserving memory for the specific list contexts (*Jacoby et al., 2015*; *Wahlheim and Jacoby, 2013*). To link our results to this remindings-based account of proactive and retroactive effects of memory, referred to as the memory-for-change framework, we propose that pre-existing relationships between base and supplemental pairs increased the likelihood of remindings, which resulted in retrieval practice benefits to more existing memories and provided more opportunities to integrate them with changed associations

during new learning. This account could be verified in future studies including manipulations of semantic relatedness along with overt measures of noticing and recollecting changes.

We next consider these results in more detail for each condition. In the ΔTarget condition, we found overall 48 hr benefits (relative to the Control condition) but no correlation with target relatedness using the narrower stimulus set. Conversely, we found no overall long-term benefits using the wider stimulus set, but we did find correlations between target relatedness and memorability. We note the surprising result of a non-significant correlation between relatedness and memorability using the narrower stimulus set, but the analyses combining narrower and wider stimulus sets at a 48 hr delay produced a positive correlation. No PF occurred in the 5 min experiments, but we found overall PI using the wider stimulus set (*Figure 6—figure supplement 1*). Altogether, these results showed that PI can still occur with low relatedness, like in other studies finding PI in ΔTarget (A-B, A-D) paradigms (for a review, see *Anderson and Neely, 1996*), but PF occurs with higher relatedness. In fact, the absence of low relatedness pairs in the narrower stimulus set likely led to the strong overall PF in this condition across all pairs (positive y-intercept in the upper right of *Figure 3A*). In this instance, there may have been a stronger influence of a binary factor (whether they are related or not), though this remains speculative and is not the case for other analyses in our paper. We believe these effects arise from the competing processes of impairments between competing responses at retrieval that have not been integrated versus retrieval benefits when that integration has occurred (which occurs especially often with high target relatedness). These types of competing processes appear operative in various associative learning paradigms such as retrieval-induced forgetting (*Anderson and McCulloch, 1999*; *Carroll et al., 2007*), and the fan effect (*Moeser, 1979*; *Reder and Anderson, 1980*). This is exactly the proposal of the memory-for-change framework, which has received support from studies showing that PF occurs when changes (integrated representations) are recollected and PI when such changes are not recollected (for a review, see *Wahlheim et al., 2021*).

In the ΔCue condition, we generally found PF in every experiment. Moreover, PF increased with cue relatedness at a 48 hr delay using the wider stimulus set; the only PI in any experiment was for low relatedness items using the wider stimulus set at a 48 hr delay, wherein memorability values from pairs (averaged across subjects) in the lowest third of relatedness were recalled worse than the Control condition. These results are consistent with Osgood's observation that interference tends to be absent or low when targets remain constant and cues change as opposed to when targets change and cues remain constant (e.g. the ΔTarget condition). The general facilitation and lack of interference effects are consistent with the idea that each independent cue does not become overloaded and can elicit the same (and only) target with which it was paired (*Anderson and Neely, 1996*; *Antony et al., 2022b*; *Watkins and Watkins, 1975*). Additionally, repeating the target, even with another cue, may increase its accessibility (at minimum) or evoke a reminding, leading to overall benefits (*Estes, 1979*; *Martin, 1965*). The prior literature on results for this condition has not been particularly reliable or theoretically extensive; we speculate that the selective PI we see could arise via confusion over whether the same target was indeed assigned twice or due to a very weak form of interference involving weight changes after targets become assigned to a new memory. Benefits could indicate that subjects could mentally link Δcue → cue → target, which would link both cues together via the reminding event; the overall correlations between relatedness and interdependence support this idea. In sum, the ΔCue condition largely resulted in more consistent PF than the ΔTarget condition, perhaps because benefits arose via the same mechanism (remindings-enabled interdependence) or even enhanced target response availability while not suffering the same drawbacks of competition that occurs when multiple non-integrated targets are linked to the same cue.

Next, we will discuss proactive effects in the ΔBoth condition, for which the prior literature is virtually non-existent. In the narrower stimulus set, 48 hr delay experiment, we found overall PF effects. These findings resemble those from our prior retroactive study (*Antony et al., 2022b*), such that this was the only experiment in which we found overall memory effects. However, unlike that prior study, here there was no correlation between cue +target relatedness and memorability. These findings also resemble general benefits of relatedness in other conditions and experiments, with the difference being that changes to both cues and targets mean that both relatedness values need to be relatively high for remindings (and their associated strengthening) to occur. That is, it is much easier for a subject to consider a new pair with both a cue related to an old cue and a target related to an old target to simply be a new word pair, causing fewer reminders overall. Interestingly, in three of

the experiments, we found significant correlations between cue +target relatedness and interdependence. It is less clear why we found interdependence in some experiments with an absence of memory boost; we speculate that in some cases, subjects may recognize the link between related pairs in the ΔBoth condition only at final retrieval.

Putting these conditions together, we consider our findings to largely support Osgood's key ideas. That is, Osgood had the bold idea that the memory surface predicting facilitation vs. interference applied in both the retroactive and proactive directions. We propose that semantic relatedness, by increasing the likelihood of remindings, provides a mechanism by which existing memories can become more accessible and associated with new learning as part of integrated memory representations that preserve targets and their contexts. Specifically, along both the cue and target identity lines of Osgood's surface, the increasing proactive benefits strongly support his proposal that PI becomes PF with high enough relatedness. However, he did not predict PI in the ΔCue condition, and additionally, we found weak support for PF inching towards the cue and target identity point along the 2-D part of the surface in the ΔBoth condition (which was present in the retroactive narrower stimulus set, 48 hr experiment of our prior paper).

While Osgood predicted that retroactive and proactive memory effects worked in both directions, he never explicitly stated whether they should be symmetric. By showing stronger retroactive than proactive benefits here, we propose that the retroactive benefits are greater, presumably because target responses receive retrieval practice during remindings. Indeed, there are recent similar results in between-subjects comparisons of retroactive and proactive memory effects in A-B, A-D paradigms (*Wahlheim et al., 2024*). However, our assertion awaits additional empirical verification using experiments that directly compare retroactive and proactive effects within-subjects and at various retention intervals. Additionally, these effects partially resemble those on spontaneous recovery, whereby original associations tend to face interference after new, conflicting learning, but slowly recover over time (either absolutely or relative to the new learning) and often eventually eclipse memory for the new information (*Barnes and Underwood, 1959*; *Postman et al., 1969*; *Wheeler, 1995*). In both cases, original associations appear more robust to change over time, though it is unclear whether these similar outcomes stem from similar mechanisms.

Our experiment had a few limitations. First, although we showed highly consistent results across retroactive and proactive comparisons and used the same subject populations and procedure, these comparisons across experiments did not involve random assignment (i.e. all experiments assessing retroactive comparisons were completed before those assessing proactive comparisons). Second, we largely explained our results using remindings-based accounts, but we did not directly manipulate remindings or measure such remindings during new learning (*Jacoby et al., 2015*). These limitations could be considered in future research. Yet another promising future direction would be to investigate how learning in this paradigm affects different aspects of the memory trace, either by investigating perceptual versus conceptual changes (*Lifanov et al., 2021*; *Melega and Sheldon, 2023*) or the semantic representations themselves (*Walsh and Rissman, 2023*). Finally, future studies could ensure these effects are not limited to these stimuli and generalize to other word stimuli in addition to testing other domains (*Baek and Papaj, 2024*; *Holding, 1976*).

Our goal in this paper was to examine fundamental questions about learning and memory for related pieces of information. Our prior work examined the retroactive impact of relatedness on existing memories (*Antony et al., 2022b*), whereas, here, we examined how relatedness proactively affected memory for new learning. Both studies were inspired by Osgood's proposal that relatedness facilitates memory strength in both directions – and both largely support his proposal. Here we offer support to the idea that, by increasing the likelihood of remindings, both old and new information can mutually benefit from the newly integrated memories, so long as they promote later recollection. Nevertheless, because retrieved memories enjoy the benefits of retrieval practice, retroactive effects appear to produce stronger benefits for memories recalled at a 48 hr delay. Therefore, to return to the example about vermouth from the introduction, if one first learns to make a martini using dry vermouth and then, while learning about making Manhattans using sweet vermouth, they recall the martini ingredients, both memories should benefit from this mental integration, but the martini memory could benefit more. We believe these findings have important ramifications for how linking old and new associations can influence human decision-making (*Nicholas et al., 2023*; *Shohamy and Daw, 2015*; *Wimmer and Shohamy, 2012*) and

reinforce and clarify the significance of building up complex networks of associations for learning and memory theory.

## Methods

### Subjects

We collected data online from 200 subjects in each experiment to amply power memorability measures from each condition. We used a five-way counterbalance, indicating that there would be 200/5 = 40 data points going into each memorability value. All subjects were undergraduate students who received psychology course credit for participating and self-attested to having normal or corrected-to-normal vision. We excluded subjects with memory performance <4 standard deviations below the mean; we ran additional subjects until again reaching 200 without exclusions. Additionally, numerous subjects did not return or complete the final test. The final breakdowns were as follows: narrower stimulus set, 5 min delay: N=212, 127 female, 3 non-binary; narrower stimulus set, 48 hr delay: N=205, 159 female, 0 non-binary; wider stimulus set, 5 min delay: N=203, 138 female, 3 non-binary; wider stimulus set, 48 hr delay: N=214, 141 female, 6 non-binary; flipped-order experiment: N=223, 141 female, 2 non-binary. All subjects were recruited via an online scheduling software or Google document set up for a psychology course. Subjects provided informed consent, and all procedures were in accordance with the California Polytechnic State University, San Luis Obispo Institutional Review Board proposal #2020–068 CP. Note that in the prior paper examining retroactive memory effects (*Antony et al., 2022b*), all experiments were similarly run online except for the narrower stimulus set, 48 hr delay experiment, wherein subjects used lab computers (before the COVID-19 pandemic). Note also that in that prior study, what we refer to here as 'supplemental pairs' were called 'secondary pairs'; we changed the language to avoid confusion, as these were learned first in this study. All subjects engaged in each step of the experiment with the guidance of research assistants; that is, all sessions were conducted synchronously online.

### Stimuli and relatedness metrics

We created our experiments with the memorability measure in mind to control for the actual three to five letter words recalled across individuals. Therefore, the main memories of interest (base pairs) involved the same 45 pairs for everyone within an experiment, meaning that we altered only the experimental condition of the supplemental pairs and the level of semantic relatedness between base and supplemental pairs across subjects. See *Antony et al., 2022b* for a detailed discussion of how stimuli were selected and allocated to counterbalance conditions across subjects. Note that for the narrower stimulus set, AS values are directed, which means that the strength between a given base → supplemental pair word was not equal to the same supplemental → base pair word. We designed the studies to systematically vary the AS from supplemental → base pair word, but in the prior study, this involved looking backward in time, whereas here, it involved looking forward in time. We bring attention to both directed AS values in the Results section to ensure this factor did not alone explain differences between the studies. For the wider stimulus set experiments, GloVe relationships were undirected, and therefore the base-supplemental pair decisions were arbitrary. All relatedness metrics were identical to those in *Antony et al., 2022b*.

### Procedure

The four main experiments in this paper followed this order: supplemental pair learning, base pair learning, a 5 min or 48 hr delay, base pair testing, and supplemental pair testing (*Figure 1A*). For supplemental pair learning, subjects first viewed 36 pairs followed by multiple rounds of testing. During encoding, pairs were preceded by a one-second fixation cross before appearing for four seconds. Cue and target words were shown centrally in the horizontal dimension and just above and below the vertical center of the screen, respectively. During retrieval-to-criterion testing, a one-second fixation cross preceded the appearance of the cue word; one second after the cue word, a blank prompt appeared in which subjects were asked to type in the target response. Subjects had no time limits for their response, and both cue and target words were shown as feedback after the response. Correct pairs were dropped from later rounds of testing until all pairs were correctly retrieved once. We did not use spell checking during learning, meaning that in some cases pairs could have been

essentially retrieved more than once. However, we do not believe this would differ across conditions to affect learning results.

Before base pair learning, subjects were told they would learn a new list of pairs that may or may not change between lists. Then base pair learning proceeded identically to supplemental pair learning, except that 45 rather than 36 pairs were learned.

At final test, we asked subjects to first recall all words from the base list (which we called the second list). Test trials involved a one-second fixation cross followed by the cue word and unlimited time to respond. Subjects received no feedback during this test. Following base pair testing, we asked subjects to recall all words from the supplemental list (which we called the first list), which had the same format. Finally, subjects were debriefed and allowed to leave.

## Statistics

In each experiment, comparisons across conditions used one-way (condition: No Δ, ΔTarget, ΔCue, ΔBoth, Control), repeated-measures ANOVAs. When significant, these were followed up using pair-wise, within-subject, FDR-corrected (*Benjamini and Hochberg, 1995*) t-tests.

Memorability, memory dependence, and bootstrapping analyses were performed identically to our prior paper (*Antony et al., 2022b*), as were methods for creating scatterplots and 2-D and 3-D surface plots. Briefly, for the memorability correlation analyses, we calculated the proportion of subjects recalling each base pair for each experimental condition minus the proportion in the Control condition. Next, we ran linear regression between the experimental – Control memorability of that pair and its relatedness value. In the ΔBoth condition, we added cue +target relatedness values together. The *y*-intercept of the ordinary least squares line was theoretically meaningful in some analyses – that is, 0 AS means 0 subjects might endorse a word in a free association task, but 0 GloVe cosine similarity is not necessarily more meaningful than other low GloVe values. The slope indicated the relationship between relatedness and memorability. Two MATLAB functions were used in the memorability plots. The slope and intercept were added to each plot based on p values from 'fitlm', and best-fit lines were added along with the confidence error output from 'polypredci' (*Strider, 2021*).

Across-experiment memory comparisons involved subtracting the No Δ from all other conditions and running between-subject corrected *t*-tests within the same condition. Results from these tests were corrected for multiple comparisons using FDR correction.

## Acknowledgements

The authors would especially like to thank Rebecca Slagle Luenser and Anthony Vierra for their work coordinating data collection on these experiments, as well as Valency Jarvis and Rachel Saconi for extensive data collection efforts across multiple terms. The authors would also like to thank the following individuals (all Cal Poly undergraduate students) for running participants across the five experiments: Nicole Attridge, Leigh Ann Bardman, Nicole Brault, Kylie Capella, Priyanka Chandar, Macey Coffman, Sadie Cooper, Rasha Demeter, Caprice Depetro, Chloe Fleischer, Daniel Frazier, Samantha Garrett, Trevor Guerra, Olivia Harrington, McKenzie Harrison, Amanda Hill, Erika Holloway, Sarah Howie, An Huynh, Rebecca Iniguez, Katie Johansen, Emma Klebaner, Carlo Lopiccolo, Jayelin Lombard, Angelo Lozano, Corinne Lykins, Rachel Nebel, Jordyn Niemiec, Sahar Oliaei, Suhana Patel, Katherine Preston, Pilar Reyes, Hector Reyes, Elli Rouse, Macy Rowe, Nami Saito, Lily Sanz, Michael Scarpa, Colin Schmitt, Christina Schwake, Evan Schweitzer, Jessica Shaver, Michela Smith, Luke Smith, Isabella Strawn, Aly Tierney, Sarah Tung, Jacob Van Dam, Mia Venturini, and Chase Volz. No external funding was received for this work.

## Additional information

### Funding
No external funding was received for this work.

## Author contributions
Kelly A Bennion, Conceptualization, Resources, Formal analysis, Supervision, Methodology, Writing – original draft, Project administration, Writing – review and editing; Jade Phong, Mytien Le, Kunhua Cheng, Supervision, Project administration; Christopher N Wahlheim, Writing – review and editing; James W Antony, Conceptualization, Data curation, Formal analysis, Supervision, Visualization, Methodology, Writing – original draft, Writing – review and editing

## Author ORCIDs
Kelly A Bennion ⓘ https://orcid.org/0000-0002-3301-7744

## Ethics
All subjects were recruited via an online scheduling software or Google document set up for a psychology course. Subjects provided informed consent, and all procedures were in accordance with the California Polytechnic State University, San Luis Obispo Institutional Review Board proposal #2020-068-CP.

Reviewer #1 (Public review): https://doi.org/10.7554/eLife.95480.3.sa1
Reviewer #2 (Public review): https://doi.org/10.7554/eLife.95480.3.sa2
Reviewer #3 (Public review): https://doi.org/10.7554/eLife.95480.3.sa3
Author response https://doi.org/10.7554/eLife.95480.3.sa4

---

# Additional files

## Supplementary files
• MDAR checklist

## Data availability
All code and data related to this paper are available at https://osf.io/u29hy/.

The following dataset was generated:

| Author(s) | Year | Dataset title | Dataset URL | Database and Identifier |
|---|---|---|---|---|
| Antony JW | 2024 | Semantic relatedness retroactively and proactively increases learning, memory, and interdependence across episodes | https://osf.io/u29hy/ | Open Science Framework, u29hy |

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
