## [Editor Report · eLife Assessment]

This **important** work advances our understanding of how memories interact to facilitate or interfere with one another, also informing our understanding of how humans build knowledge. The study provides **compelling** evidence that semantic relatedness proactively benefits memory using clean experimental design, rigorous statistics, large N samples, and well-characterized stimuli. The study also demonstrates the boundaries of these proactive benefits, showing that when studied items have weaker semantic relationships, proactive interference may be observed. This research will be of interest to memory researchers as well as cognitive psychologists, neuroscientists, and educators more broadly.

---

## [Referee Report · Reviewer #1 (Public review)]

Summary:

Bennion and colleagues present a careful examination of how an earlier set of memories can either interfere with or facilitate memories formed later. This impressive work is a companion piece to an earlier paper by Antony and colleagues (2022) in which a similar experimental design was used to examine how a later set of memories can either interfere with or facilitate memories formed earlier. This study makes contact with an experimental literature spanning 100 years, which is concerned with the nature of forgetting, and the ways in which memories for particular experiences can interact with other memories. These ideas are fundamental to modern theories of human memory, for example, paired-associates studies like this one are central to the theoretical idea that interference between memories is a much bigger contributor to forgetting than any sort of passive decay.

Strengths:

At the heart of the current investigation is a proposal made by Osgood in the 1940s regarding how paired associates are learned and remembered. In these experiments one learns a pair of items, A-B (cue-target), and then later learns another pair that is related in some way, either A'-B (changing the cue, delta-cue), or A-B' (changing the target, delta-target), or A'-B' (changing both, delta-both), where the prime indicates that item has been modified, and may be semantically related to the original item. The authors refer to the critical to-be-remembered pairs as base pairs. Osgood proposed that when the changed item is very different from the original item there will be interference, and when the changed item is similar to the original item there will be facilitation. Osgood proposed a graphical depiction of his theory in which performance was summarized as a surface, with one axis indicating changes to the cue item of a pair and the other indicating changes to the target item, and the surface itself necessary to visualize the consequences of changing both.

In the decades since Osgood's proposal, there have been many studies examining slivers of the proposal, e.g., just changing targets in one experiment, just changing cues in another experiment. Because any pair of experiments use different methods, this has made it difficult to draw clear conclusions about the effects of particular manipulations.

The current paper is a potential landmark, in that they manipulate multiple fundamental experimental characteristics using the same general experimental design. Importantly, they manipulate the semantic relatedness of the changed item to the original item, the delay between the study experience and the test, and which aspect of the pair is changed. Furthermore, they include both a positive control condition (where the exact same pair is studied twice), and a negative control condition (where a pair is only studied once, in the same phase as the critical base pairs). This allows them to determine when the prior learning exhibits an interfering effect relative to the negative control condition, and also allows them to determine how close any facilitative effects come to matching the positive control.

The results are interpreted in terms of a set of existing theories, most prominently the memory-for-change framework, which proposes a mechanism (recursive reminding) potentially responsible for the facilitative effects examined here. One of the central results is the finding that a stronger semantic relationship between a base pair and an earlier pair has a facilitative effect on both the rate of learning of the base pair and the durability of the memory for the base pair. This is consistent with the memory-for-change framework, which proposes that this semantic relationship prompts retrieval of the earlier pair, and the two pairs are integrated into a common memory structure that contains information about which pair was studied in which phase of the experiment. When semantic relatedness is lower, they more often show interference effects, with the idea being that competition between the stored memories makes it more difficult to remember the base pair.

This work represents a major methodological and empirical advance for our understanding of paired-associates learning, and it sets a laudably high bar for future work seeking to extend this knowledge further. By manipulating so many factors within one set of experiments, it fills a gap in the prior literature regarding the cognitive validity of an 80-year-old proposal by Osgood. The reader can see where the observed results match Osgood's theory and where they are inconclusive. This gives us insight, for example, into the necessity of including a long delay in one's experiment, to observe potential facilitative effects. This point is theoretically interesting, but it is also a boon for future methodological development, in that it establishes the experimental conditions necessary for examining one or another of these facilitation or interference effects more closely.

The authors were exceptionally responsive to the suggestions of the reviewers, and the revisions have improved the theoretical clarity of the paper. I think the value of this work will grow with time, as memory researchers and theorists use it as a benchmark for new theory development. For example, the data from these experiments will undoubtedly be used to develop and constrain a new generation of computational models of paired-associates learning.

Weaknesses:

One minor weakness of the work is that the overarching theoretical framing does not necessarily specify the expected result for each and every one of the many effects examined. For example, with a narrower set of semantic associations being considered (all of which are relatively high associations) and a long delay, varying the semantic relatedness of the target item did not reliably affect the memorability of that pair. However, the same analysis showed a significant effect when the wider set of semantic associations was used. The positive result is consistent with the memory-for-change framework, but the null result isn't clearly informative to the theory. However, research is never done; comparing the results with the two sets of semantic associations is informative from a methodological perspective, in that it establishes the degree to which semantic relatedness must be altered to affect behavioral performance in a paired-associates task.

---

## [Referee Report · Reviewer #2 (Public review)]

Summary:

The study focuses on how relatedness with existing memories affects the formation and retention of new memories. Of core interest were the conditions that determine when prior memories facilitate new learning or interfere with it. Across a set of experiments that varied the degree of relatedness across memories as well as retention interval, the study compellingly shows that relatedness typically leads to proactive facilitation of new learning, with interference only observed under specific conditions and immediate test and being thus an exception rather than a rule.

Strengths:

The study uses a well-established word-pair learning paradigm to study interference and facilitation of overlapping memories. It however goes more in depth than a typical interference study in the systematic variation of several factors: (1) which elements of an association are overlapping and which are altered (change target, change cue, change both, change neither); (2) how much the changed element differs from the original (word relatedness, with two ranges of relatedness considered); (3) retention period (immediate test, 2-day delay). Furthermore, each experiment has a large N sample size, so both significant effects as well as null effects are robust and informative.

The results show the benefits of relatedness, but also replicate interference effects in the "change target" condition when the new target is not related to the old target and when test is immediate. This provides reconciliation of some existing seemingly contradictory results on the effect of overlap on memory. Here, the whole range of conditions is mapped to convincingly show how the direction of the effect can flip across the surface of relatedness values.

Additional strength comes from supporting analyses, such as analyses of learning data, demonstrating that relatedness leads to both better final memory and also faster initial learning.

More broadly, the study informs our understanding of memory integration, demonstrating how interdependence of memory for related information increases with relatedness. Together with a prior study or retroactive interference and facilitation, the results provide new insights into the role of reminding in memory formation.

In summary, this is a highly rigorous body of work that sets a great model for future studies and improves our understanding of memory organization.

Weaknesses:

The evidence for the proactive facilitation driven by relatedness is very convincing. However, in the finer scale results, the continuous relationship between the degree of relatedness and the degree of proactive facilitation/interference is less clear. The relationship was only found in the wider stimulus set, where some pairs were unrelated and other pairs related, and only when GloVe metric for measuring relatedness was used. The absence of a relationship between relatedness and memory in the narrow stimulus set (where all pairs were related to some degree) suggests this could be potentially an all-or-none effect (facilitation for related) rather than a matter of degree. Furthermore, a different metric of relatedness, associative strength AS, did not show the same relationship. The discrepancy between the metrics is not fully resolved. This is less of a problem with interdependence analyses where the results are more converging across narrow and wider range as well as the two metrics.

A smaller weakness, acknowledged by the authors, is generalizability beyond the word set used here. Using a carefully crafted stimulus set and repeating the same word pairings across participants and conditions was important for memorability calculations and some of the other analyses. However, highlighting the inherently noisy item-by-item results, especially in the Osgood-style surface figures, makes it challenging to imagine how the results would generalize to new stimuli, even within the same relatedness ranges as the current stimulus sets.

---

## [Referee Report · Reviewer #3 (Public review)]

Summary:

Bennion et al. investigate how semantic relatedness proactively benefits the learning of new word pairs. The authors draw predictions from Osgood (1949), which posits that the degree of proactive interference (PI) and proactive facilitation (PF) of previously learned items on to-be-learned items depends on the semantic relationships between the old and new information. In the current study, participants subjects learn a set of word pairs ("supplemental pairs"), followed by a second set of pairs ("base pairs"), in which the cue, target or both words are changed, or the pair was identical. Pairs were drawn from either a narrower or wider stimulus set and were tested after either a 5 minute or 48 hour delay. The results show that semantic relatedness overwhelmingly produces PF and greater memory interdependence between base and supplemental pairs, except in the case of unrelated pairs in a wider stimulus set after a short delay, which produced PI. In their final analyses, the authors compare their current results to previous work from their group studying the analogous retroactive effects of semantic relatedness on memory. These comparisons show generally similar, if slightly weaker, patterns of results. The authors interpret their results in the framework of recursive reminders (Hintzman, 2011), which posits that the semantic relationships between new and old word pairs promotes reminders of the old information during the learning of the new to-be-learned information. These reminders help to integrate the old and new information and result in additional retrieval practice opportunities that in turn improve later recall.

Strengths:

Overall, I thought that the analyses were thorough and well-thought-out and the results were incredibly well-situated in the literature, especially with the additional clarification and framing that the authors have made in response to reviewer comments. In particular, I found that the large sample size, inclusion of a wide range of semantic relatedness across the two stimulus sets, variable delays and the ability to directly compare the current results to their prior results on the retroactive effects of semantic relatedness were particular strengths of the authors' approach and make this an impressive contribution to the existing literature. I thought that their interpretations and conclusions were mostly reasonable and included appropriate caveats (where applicable).

Weaknesses:

The changes and additional analyses that the authors have made have addressed my concerns about their analyses. Including the additional Fig 1- Supp 1, panel C greatly helps with the interpretability across stimulus sets, and the additional analyses the authors have performed teasing apart whether cue and target similarity separately influence memorability and interdependence seem to support the rest of their conclusions.

---

## [Author Response]

The following is the authors’ response to the original reviews.

**Public Reviews:**

**Reviewer #1 (Public Review):**
Summary:Bennion and colleagues present a careful examination of how an earlier set of memories can either interfere with or facilitate memories formed later. This impressive work is a companion piece to an earlier paper by Antony and colleagues (2022) in which a similar experimental design was used to examine how a later set of memories can either interfere with or facilitate memories formed earlier. This study makes contact with an experimental literature spanning 100 years, which is concerned with the nature of forgetting, and the ways in which memories for particular experiences can interact with other memories. These ideas are fundamental to modern theories of human memory, for example, paired-associate studies like this one are central to the theoretical idea that interference between memories is a much bigger contributor to forgetting than any sort of passive decay.Strengths:At the heart of the current investigation is a proposal made by Osgood in the 1940s regarding how paired associates are learned and remembered. In these experiments, one learns a pair of items, A-B (cue-target), and then later learns another pair that is related in some way, either A'-B (changing the cue, delta-cue), or A-B' (changing the target, delta-target), or A'-B' (changing both, delta-both), where the prime indicates that item has been modified, and may be semantically related to the original item. The authors refer to the critical to-be-remembered pairs as base pairs. Osgood proposed that when the changed item is very different from the original item there will be interference, and when the changed item is similar to the original item there will be facilitation. Osgood proposed a graphical depiction of his theory in which performance was summarized as a surface, with one axis indicating changes to the cue item of a pair and the other indicating changes to the target item, and the surface itself necessary to visualize the consequences of changing both.In the decades since Osgood's proposal, there have been many studies examining slivers of the proposal, e.g., just changing targets in one experiment, just changing cues in another experiment. Because any pair of experiments uses different methods, this has made it difficult to draw clear conclusions about the effects of particular manipulations.The current paper is a potential landmark, in that the authors manipulate multiple fundamental experimental characteristics using the same general experimental design. Importantly, they manipulate the semantic relatedness of the changed item to the original item, the delay between the study experience and the test, and which aspect of the pair is changed. Furthermore, they include both a positive control condition (where the exact same pair is studied twice), and a negative control condition (where a pair is only studied once, in the same phase as the critical base pairs). This allows them to determine when the prior learning exhibits an interfering effect relative to the negative control condition and also allows them to determine how close any facilitative effects come to matching the positive control.The results are interpreted in terms of a set of existing theories, most prominently the memory-for-change framework, which proposes a mechanism (recursive reminding) potentially responsible for the facilitative effects examined here. One of the central results is the finding that a stronger semantic relationship between a base pair and an earlier pair has a facilitative effect on both the rate of learning of the base pair and the durability of the memory for the base pair. This is consistent with the memory-for-change framework, which proposes that this semantic relationship prompts retrieval of the earlier pair, and the two pairs are integrated into a common memory structure that contains information about which pair was studied in which phase of the experiment. When semantic relatedness is lower, they more often show interference effects, with the idea being that competition between the stored memories makes it more difficult to remember the base pair.This work represents a major methodological and empirical advance for our understanding of paired-associates learning, and it sets a laudably high bar for future work seeking to extend this knowledge further. By manipulating so many factors within one set of experiments, it fills a gap in the prior literature regarding the cognitive validity of an 80-year-old proposal by Osgood. The reader can see where the observed results match Osgood's theory and where they are inconclusive. This gives us insight, for example, into the necessity of including a long delay in one's experiment, to observe potential facilitative effects. This point is theoretically interesting, but it is also a boon for future methodological development, in that it establishes the experimental conditions necessary for examining one or another of these facilitation or interference effects more closely.

We thank the reviewer for their thorough and positive comments -- thank you so much!

Weaknesses:One minor weakness of the work is that the overarching theoretical framing does not necessarily specify the expected result for each and every one of the many effects examined. For example, with a narrower set of semantic associations being considered (all of which are relatively high associations) and a long delay, varying the semantic relatedness of the target item did not reliably affect the memorability of that pair. However, the same analysis showed a significant effect when the wider set of semantic associations was used. The positive result is consistent with the memory-for-change framework, but the null result isn't clearly informative to the theory. I call this a minor weakness because I think the value of this work will grow with time, as memory researchers and theorists use it as a benchmark for new theory development. For example, the data from these experiments will undoubtedly be used to develop and constrain a new generation of computational models of paired-associates learning.

We thank the reviewer for this constructive critique. We agree that the experiments with a narrower set of semantic associations are less informative; in fact, we thought about removing these experiments from the current study, but given that we found results in the ΔBoth condition in Antony et al. (2022) using these stimuli that we did NOT find in the wider set, we thought it was worth including for a thorough comparison. We hope that the analyses combining the two experiment sets (Fig 6-Supp 1) are informative for contextualizing the results in the ‘narrower’ experiments and, as the reviewer notes, for informing future researchers.

**Reviewer #2 (Public Review):**
Summary:The study focuses on how relatedness with existing memories affects the formation and retention of new memories. Of core interest were the conditions that determine when prior memories facilitate new learning or interfere with it. Across a set of experiments that varied the degree of relatedness across memories as well as retention interval, the study compellingly shows that relatedness typically leads to proactive facilitation of new learning, with interference only observed under specific conditions and immediate test and being thus an exception rather than a rule.Strengths:The study uses a well-established word-pair learning paradigm to study interference and facilitation of overlapping memories. However it goes more in-depth than a typical interference study in the systematic variation of several factors: (1) which elements of an association are overlapping and which are altered (change target, change cue, change both, change neither); (2) how much the changed element differs from the original (word relatedness, with two ranges of relatedness considered); (3) retention period (immediate test, 2-day delay). Furthermore, each experiment has a large N sample size, so both significant effects as well as null effects are robust and informative.The results show the benefits of relatedness, but also replicate interference effects in the "change target" condition when the new target is not related to the old target and when the test is immediate. This provides a reconciliation of some existing seemingly contradictory results on the effect of overlap on memory. Here, the whole range of conditions is mapped to convincingly show how the direction of the effect can flip across the surface of relatedness values.Additional strength comes from supporting analyses, such as analyses of learning data, demonstrating that relatedness leads to both better final memory and also faster initial learning.More broadly, the study informs our understanding of memory integration, demonstrating how the interdependence of memory for related information increases with relatedness. Together with a prior study or retroactive interference and facilitation, the results provide new insights into the role of reminding in memory formation.In summary, this is a highly rigorous body of work that sets a great model for future studies and improves our understanding of memory organization.

We thank their reviewer for their thorough summary and very supportive words!

Weaknesses:The evidence for the proactive facilitation driven by relatedness is very convincing. However, in the finer scale results, the continuous relationship between the degree of relatedness and the degree of proactive facilitation/interference is less clear. This could be improved with some additional analyses and/or context and discussion. In the narrower range, the measure used was AS, with values ranging from 0.03-0.98, where even 0.03 still denotes clearly related words (pious - holy). Within this range from "related" to "related a lot", no relationship to the degree of facilitation was found. The wider range results are reported using a different scale, GloVe, with values from -0.14 to 0.95, where the lower end includes unrelated words (sap - laugh). It is possible that any results of facilitation/interference observed in the wider range may be better understood as a somewhat binary effect of relatedness (yes or no) rather than the degree of relatedness, given the results from the narrower condition. These two options could be more explicitly discussed. The report would benefit from providing clearer information about these measures and their range and how they relate to each other (e.g., not a linear transformation). It would be also helpful to know how the values reported on the AS scale would end up if expressed in the GloVe scale (and potentially vice-versa) and how that affects the results. Currently, it is difficult to assess whether the relationship between relatedness and memory is qualitative or quantitative. This is less of a problem with interdependence analyses where the results converge across a narrow and wider range.

We thank the reviewer for this point. While other analyses do show differences across the range of AS values we used, we agree in the case of the memorability analysis in the narrower stimulus set, 48-hr experiment (or combining across the narrower and wider stimulus sets), there could be a stronger influence of binary (yes/no) relatedness. We have now made this point explicitly (p. 26):

“Altogether, these results show that PI can still occur with low relatedness, like in other studies finding PI in ΔTarget (A-B, A-D) paradigms (for a review, see Anderson & Neely, 1996), but PF occurs with higher relatedness. In fact, the absence of low relatedness pairs in the narrower stimulus set likely led to the strong overall PF in this condition across all pairs (positive y-intercept in the upper right of Fig 3A). In this particular instance, there may have been a stronger influence of a binary factor (whether they are related or not), though this remains speculative and is not the case for other analyses in our paper.”

Additionally, we have also emphasized that the two relatedness metrics are not linear transforms of each other. Finally, as in addressing both your and reviewer #3’s comment below, we now graph relatedness values under a common GloVe metric in Fig 1-Supp 1C (p. 9):

“Please note that GloVe is an entirely different relatedness metric and is not a linear transformation of AS (see Fig 1-Supp 1C for how the two stimulus sets compare using the common GloVe metric).”

A smaller weakness is generalizability beyond the word set used here. Using a carefully crafted stimulus set and repeating the same word pairings across participants and conditions was important for memorability calculations and some of the other analyses. However, highlighting the inherently noisy item-by-item results, especially in the Osgood-style surface figures, makes it challenging to imagine how the results would generalize to new stimuli, even within the same relatedness ranges as the current stimulus sets.

We thank the reviewer for this critique. We have added this caveat in the limitations to suggest that future studies should replicate these general findings with different stimulus sets (p. 28):

“Finally, future studies could ensure these effects are not limited to these stimuli and generalize to other word stimuli in addition to testing other domains (Baek & Papaj, 2024; Holding, 1976).”

**Reviewer #3 (Public Review):**
Summary:Bennion et al. investigate how semantic relatedness proactively benefits the learning of new word pairs. The authors draw predictions from Osgood (1949), which posits that the degree of proactive interference (PI) and proactive facilitation (PF) of previously learned items on to-be-learned items depends on the semantic relationships between the old and new information. In the current study, participants learn a set of word pairs ("supplemental pairs"), followed by a second set of pairs ("base pairs"), in which the cue, target, or both words are changed, or the pair is identical. Pairs were drawn from either a narrower or wider stimulus set and were tested after either a 5-minute or 48-hour delay. The results show that semantic relatedness overwhelmingly produces PF and greater memory interdependence between base and supplemental pairs, except in the case of unrelated pairs in a wider stimulus set after a short delay, which produced PI. In their final analyses, the authors compare their current results to previous work from their group studying the analogous retroactive effects of semantic relatedness on memory. These comparisons show generally similar, if slightly weaker, patterns of results. The authors interpret their results in the framework of recursive reminders (Hintzman, 2011), which posits that the semantic relationships between new and old word pairs promote reminders of the old information during the learning of the new to-be-learned information. These reminders help to integrate the old and new information and result in additional retrieval practice opportunities that in turn improve later recall.Strengths:Overall, I thought that the analyses were thorough and well-thought-out and the results were incredibly well-situated in the literature. In particular, I found that the large sample size, inclusion of a wide range of semantic relatedness across the two stimulus sets, variable delays, and the ability to directly compare the current results to their prior results on the retroactive effects of semantic relatedness were particular strengths of the authors' approach and make this an impressive contribution to the existing literature. I thought that their interpretations and conclusions were mostly reasonable and included appropriate caveats (where applicable).

We thank the reviewer for this kind, effective summary and highlight of the paper’s strengths!

Weaknesses:Although I found that the paper was very strong overall, I have three main questions and concerns about the analyses.My first concern lies in the use of the narrow versus wider stimulus sets. I understand why the initial narrow stimulus set was defined using associative similarity (especially in the context of their previous paper on the retroactive effects of semantic similarity), and I also understand their rationale for including an additional wider stimulus set. What I am less clear on, however, is the theoretical justification for separating the datasets. The authors include a section combining them and show in a control analysis that there were no directional effects in the narrow stimulus set. The authors seem to imply in the Discussion that they believe there are global effects of the lower average relatedness on differing patterns of PI vs PF across stimulus sets (lines 549-553), but I wonder if an alternative explanation for some of their conflicting results could be that PI only occurs with pairs of low semantic relatedness between the supplemental and base pair and that because the narrower stimulus set does not include the truly semantically unrelated pairs, there was no evidence of PI.

We agree with the reviewer’s interpretation here, and we have now directly stated this in the discussion section (p. 26):

“Altogether, these results show that PI can still occur with low relatedness, like in other studies finding PI in ΔTarget (A-B, A-D) paradigms (for a review see, Anderson & Neely, 1996), but PF occurs with higher relatedness. In fact, the absence of low relatedness pairs in the narrower stimulus set likely led to the strong overall PF in this condition across all pairs (positive y-intercept in the upper right of Fig 3A).”

As for the remainder of this concern, please see our response to your elaboration on the critique below.

My next concern comes from the additive change in both measures (change in Cue + change in Target). This measure is simply a measure of overall change, in which a pair where the cue changes a great deal but the target doesn't change is treated equivalently to a pair where the target changes a lot, but the cue does not change at all, which in turn are treated equivalently to a pair where the cue and target both change moderate amounts. Given that the authors speculate that there are different processes occurring with the changes in cue and target and the lack of relationship between cue+target relatedness and memorability, it might be important to tease apart the relative impact of the changes to the different aspects of the pair.

We thank the reviewer for this great point. First, we should clarify that we only added cue and target similarity values in the ΔBoth condition, which means that all instances of equivalence relate to non-zero values for both cue and target similarity. However, it is certainly possible cue and target similarity separately influence memorability or interdependence. We have now run this analysis separately for cue and target similarity (but within the ΔBoth condition). For memorability, neither cue nor target similarity independently predicted memorability within the ΔBoth condition in any of the four main experiments (all *p* > 0.23). Conversely, there were some relationships with interdependence. In the narrower stimulus set, 48-hr delay experiment, both cue and target similarity significantly or marginally predicted base-secondary pair interdependence (Cue: *r* = 0.30, *p* = 0.04; Target: *r* = 0.29, *p* = 0.054). Notably, both survived partial correlation analyses partialing out the other factor (Cue: *r* = 0.33, *p* = 0.03; Target: *r* = 0.32, *p* = 0.04). In the wider stimulus set, 48-hr delay experiment, only target similarity predicted interdependence (Cue: *r* = 0.09, *p* = 0.55; Target: *r* = 0.34, *p* = 0.02), and target similarity also predicted interdependence after partialing out cue similarity (*r* = 0.34, *p* = 0.02). Similarly, in the narrower stimulus set, 5-min delay experiment, only target similarity predicted interdependence (Cue: *r* = 0.01, *p* = 0.93; Target: *r* = 0.41, *p* = 0.005), and target similarity also predicted interdependence after partialing out cue similarity (*r* = 0.42, *p* = 0.005). Neither predicted interdependence in the wider stimulus set, 5-min delay experiment (Cue: *r* = -0.14, *p* = 0.36; Target: *r* = 0.09, *p* = 0.54). We have opted to leave this out of the paper for now, but we could include it if the reviewer believes it is worthwhile.

Note that we address the multiple regression point raised by the reviewer in the critique below.

Finally, it is unclear to me whether there was any online spell-checking that occurred during the free recall in the learning phase. If there wasn't, I could imagine a case where words might have accidentally received additional retrieval opportunities during learning - take for example, a case where a participant misspelled "razor" as "razer." In this example, they likely still successfully learned the word pair but if there was no spell-checking that occurred during the learning phase, this would not be considered correct, and the participant would have had an additional learning opportunity for that pair.

We did not use online spell checking. We agree that misspellings would be considered successful instances of learning (meaning that for those words, they would essentially have successful retrieval more than once). However, we do not have a reason to think that this would meaningfully differ across conditions, so the main learning results would still hold. We have included this in the Methods (p. 29-30):

“We did not use spell checking during learning, meaning that in some cases pairs could have been essentially retrieved more than once. However, we do not believe this would differ across conditions to affect learning results.”

**Recommendations for the authors:**

**Reviewer #1 (Recommendations For The Authors):**
In terms of the framing of the paper, I think the paper would benefit from a clearer explication of the different theories at play in the introductory section. There are a few theories being examined. Memory-for-change is described in most detail in the discussion, it would help to describe it more deliberately in the intro. The authors refer to a PI account, and this is contrasted with the memory-for-change account, but it seems to me that these theories are not mutually exclusive. In the discussion, several theories are mentioned in passing without being named, e.g., I believe the authors are referring to the fan effect when they mention the difference between delta-cue and delta-target conditions. Perhaps this could be addressed with a more detailed account of the theory underlying Osgood's predictions, which I believe arise from an associative account of paired-associates memory. Osgood's work took place when there was a big debate between unlearning and interference. The current work isn't designed to speak directly to that old debate. But it may be possible to develop the theory a bit more in the intro, which would go a long way towards scaffolding the many results for the reader, by giving them a better sense up front of the theoretical implications.

We thank the reviewer for this comment and the nudge to clarify these points. First, we have now made the memory-for-change and remindings accounts more explicit in the introduction, as well as the fact that we are combining the two in forming predictions for the current study (p. 3):

“Conversely, in favor of the PF account, we consider two main, related theories. The first is the importance of “remindings” in memory, which involve reinstating representations from an earlier study phase during later learning (Hintzman, 2011). This idea centers study-phase retrieval, which involves being able to mentally recall prior information and is usually applied to exact repetitions of the same material (Benjamin & Tullis, 2010; Hintzman et al., 1975; Siegel & Kahana, 2014; Thios & D’Agostino, 1976; Zou et al., 2023). However, remindings can occur upon the presentation of related (but not identical) material and can result in better memory for both prior and new information when memory for the linked events becomes more interdependent (Hintzman, 2011; Hintzman et al., 1975; McKinley et al., 2019; McKinley & Benjamin, 2020; Schlichting & Preston, 2017; Tullis et al., 2014; Wahlheim & Zacks, 2019). The second is the memory-for-change framework, which builds upon these ideas and argues that humans often retrieve prior experiences during new learning, either spontaneously by noticing changes from what was learned previously or by instruction (Jacoby et al., 2015; Jacoby & Wahlheim, 2013). The key advance of this framework is that recollecting changes is necessary for PF, whereas PI occurs without recollection. This framework has been applied to paradigms including stimulus changes, including common paired associate paradigms (e.g., A-B, A-D) that we cover extensively later. Because humans may be more likely to notice and recall prior information when it is more related to new information, these two accounts would predict that semantic relatedness instead promotes successful remindings, which would create PF and interdependence among the traces.”

Second, as the reviewer suggests, we were referring to the fan effect in the discussion, and we have now made that more explicit (p. 26):

“We believe these effects arise from the competing processes of impairments between competing responses at retrieval that have not been integrated versus retrieval benefits when that integration has occurred (which occurs especially often with high target relatedness). These types of competing processes appear operative in various associative learning paradigms such as retrieval-induced forgetting (Anderson & McCulloch, 1999; Carroll et al., 2007), and the fan effect (Moeser, 1979; Reder & Anderson, 1980).”

Finally, our reading of Osgood’s proposal is as an attempt to summarize the qualitative effects of the scattered literature (as of 1949) and did not discuss many theories. For this reason, we generally focus on the directional predictions relating to Osgood’s surface, but we couch it in theories proposed since then.

It strikes me that the advantage seen for items in the retroactive study compared to the proactive study is consistent with classic findings examining spontaneous recovery. These classic studies found that first-learned materials tended to recover to a level above second-learned materials as time passed. This could be consistent with the memory-for-change proposal presented in the text. The memory-for-change proposal provides a potential cognitive mechanism for the effect, here I'm just suggesting a connection that could be made with the spontaneous recovery literature.

We thank the reviewer for this suggestion. Indeed, we agree there is a meaningful point of connection here. We have added the following to the Discussion (p. 27):

“Additionally, these effects partially resemble those on spontaneous recovery, whereby original associations tend to face interference after new, conflicting learning, but slowly recover over time (either absolutely or relative to the new learning) and often eventually eclipse memory for the new information (Barnes & Underwood, 1959; Postman et al., 1969; Wheeler, 1995). In both cases, original associations appear more robust to change over time, though it is unclear whether these similar outcomes stem from similar mechanisms.”

Minor recommendationsLine 89: relative existing -> relative to existing.Line 132: "line from an unrelated and identical target" -> from an unrelated to identical target (take a look, just needs rephrasing).Line 340: (e.g. peace-shaverazor) I wasn't clear whether this was a typographical error, or whether the intent was to typographically indicate a unified representation.Line 383: effects on relatedness -> effects of relatedness.

We think the reviewer for catching these errors. We have fixed them, and for the third comment, we have clarified that we indeed meant to indicate a unified representation (p. 12):

“[e.g., peace-shaverazor (written jointly to emphasize the unification)]”

Page 24: Figure 8. I think the statistical tests in this figure are just being done between the pairs of the same color? Like in the top left panel, delta-cue pro and delta-target retro are adjacent and look equivalent, but there is no n.s. marking for this pair. Could consider keeping the connecting line between the linked conditions and removing the connecting lines that span different conditions.

Indeed, we were only comparing conditions with the same color. We have changed the connecting lines to reflect this.

Page 26 line 612: I think this is the first mention that the remindings account is referred to as the memory-for-change framework, consider mentioning this in the introduction.

Thank you – we have now mentioned this in the introduction.

Lines 627-630. Is this sentence referring to the fan effect? If so it could help the reader to name it explicitly.

We have now named this explicitly.

**Reviewer #2 (Recommendations For The Authors):**
This is a matter of personal preference, but I would prefer PI and PF spelled out instead of the abbreviations. This was also true for RI and RF which are defined early but then not used for 20 pages before being re-used again. In contrast, the naming of the within-subject conditions was very intuitive.

We appreciate this perspective. However, we prefer to keep the terms PI and PF for the sake of brevity. We now re-introduce terms that do not return until later in the manuscript.

Osgood surface in Figure 1A could be easier to read if slightly reformatted. For example, target and cue relatedness sides are very disproportional and I kept wondering if that was intentional. The z-axis could be slightly more exaggerated so it's easier to see the critical messages in that figure (e.g., flip from + to - effect along the one dimension). The example word pairs were extremely helpful.Figures 1C and 1D were also very helpful. It would be great if they could be a little bigger as the current version is hard to read.Figure 1B took a while to decipher and could use a little more anticipation in the body of the text. Any reason to plot the x-axis from high to low on this figure? It is confusing (and not done in the actual results figures). I believe the supplemental GloVe equivalent in the supplement also has a confusing x-axis.

Thank the reviewer for this feedback. We have modified Figure 1A to reduce the disproportionality and accentuate the *z*-axis changes. We have also made the text in C and D larger. Finally, we have flipped around the x-axis in B and in the supplement.

The description of relatedness values was rather confusing. It is not intuitive to accept that AS values from 0.03-0.96 are "narrow", as that seems to cover almost the whole theoretical range. I do understand that 0.03 is still a value showing relatedness, but more explanation would be helpful. It is also not clear how the GloVe values compare to the AS values. If I am understanding the measures and ranges correctly, the "narrow" condition could also be called "related only" while the "wide" condition could be called "related and unrelated". This is somewhat verbalized but could be clearer. In general, please provide a straightforward way for a reader to explicitly or implicitly compare those conditions, or even plot the "narrow" condition using both AS values and GloVe values so one can really compare narrow and wider conditions comparing apples with apples.

We thank the reviewer for this critique. First, we have now sought to clarify this in the Introduction (p. 11-12):

“Across the first four experiments, we manipulated two factors: range of relatedness among the pairs and retention interval before the final test. The narrower range of relatedness used direct AS between pairs using free association norms, such that all pairs had between 0.03-0.96 association strength. Though this encompasses what appears to be a full range of relatedness values, pairs with even low AS are still related in the context of all possible associations (e.g., pious-holy has AS = 0.03 but would generally be considered related) (Fig 1B). The stimuli using a wider range of relatedness spanned the full range of global vector similarity (Pennington et al., 2014) that included many associations that would truly be considered unrelated (Fig 1-Supp 1A). One can see the range of the wider relatedness values in Fig 1-Supp 1B and comparisons between narrower and wider relatedness values in Fig 1-Supp 1C.”

Additionally, as noted in the text above, we have added a new subfigure to Fig 1-Supp 1 that compares the relatedness values in the narrower and wider stimulus sets using the common GloVe metric.

Considering a relationship other than linear may also be beneficial (e.g., the difference between AS of 0.03 and 0.13 may not be equal to AS of .83 and .93; same with GloVe). I am assuming that AS and GloVe are not linear transforms of each other. Thus, it is not clear whether one should expect a linear (rather than curvilinear or another monotonic) relationship with both of them. It could be as simple as considering rank-order correlation rather than linear correlation, but just wanted to put this out for consideration. The linear approach is still clearly fruitful (e.g., interdependence), but limits further the utility of having both narrow and wide conditions without a straightforward way to compare them.

We thank the reviewer for this point. Indeed, AS and GloVe are not linear transforms of each other, but metrics derived from different sources (AS comes from human free associations; GloVe comes from a learned vector space language model). (We noted this in the text and in our response to your above comment.) However, we do have the ability to put all the word pairs into the GloVe metric, which we do in the Results section, “Re-assessing proactive memory and interdependence effects using a common metric”. In this analysis, we used a linear correlation that combined data sets with a similar retention interval and replicated our main findings earlier in the paper (p. 5):

“In the 48-hr delay experiment, correlations between memorability and cue relatedness in the ΔCue condition [r2(44) > 0.29, *p* < 0.001] and target relatedness in the ΔTarget condition [r2(44) = 0.2, *p* < 0.001] were significant, whereas cue+target relatedness in the ΔBoth condition was not [r2(44) = 0.01, *p* = 0.58]. In all three conditions, interdependence increased with relatedness [all r2(44) > 0.16, *p* < 0.001].”

Following the reviewer suggestion to test things out using rank order, we also re-created the combined analysis using rank order based on GloVe values rather than the raw GloVe values. The ranks now span 1-90 (because there were 45 pairs in each of the narrower and wider stimulus sets). All results qualitatively held.

**Author response image 1. sa4fig1:** Rank order results.

**Author response image 2. sa4fig2:** And the raw results in Fig 6-Supp 1 (as a reference).

**Reviewer #3 (Recommendations For The Authors):**
In regards to my first concern, the authors could potentially test whether the stimulus sets are different by specifically looking at pairs from the wider stimulus set that overlap with the range of relatedness from the narrow set and see if they replicate the results from the narrow stimulus set. If the results do not differ, the authors could simplify their results section by collapsing across stimulus sets (as they did in the analyses presented in Figure 6 - Supplementary Figure 1). If the authors opt to keep the stimulus sets separate, it would be helpful to include a version of Figure 1b/Figure 1 - Supplementary Figure 1 where the coverage of the two stimulus sets are plotted on the same figure using GloVe similarity so it is easier to interpret the results.

We have conducted this analysis in two ways, though we note that we will eventually settle upon keeping the stimulus sets separate. First, we examined memorability between the data sets by removing one pair at a time from the wider stimulus set until there was no significant difference (*p* > 0.05). We did this at the long delay because that was more informative for most of our analyses. Even after reducing the wider stimulus set, the narrow stimulus set still had significantly or marginally higher memorability in all three conditions (*p* < 0.001 for ΔCue; *p* < 0.001 for ΔTarget; *p* = 0.08 for ΔBoth). We reasoned that this was likely because the AS values still differed (all, *p* < 0.001), which would present a clear way for participants to associate words that may not be as strongly similar in vector space (perhaps due to polysemy for individual words). When we ran the analysis a different way that equated AS, we no longer found significant memorability differences (*p* = 0.13 for ΔCue; *p* = 0.50 for ΔTarget; *p* = 0.18 for ΔBoth). However, equating the two data sets in this analysis required us to drop so many pairs to equate the wider stimulus data set (because only a few only had a direct AS connection; there were 3, 5, and 1 pairs kept in the ΔCue, ΔTarget, and ΔBoth conditions) that we would prefer not to report this result.

Additionally, we now plot the two stimulus sets on the same plot (Reviewer 2 also suggested this).

In regards to my second concern, one potential way the authors could disambiguate the effects of change in cue vs change in target might be to run a multiple linear regression with change in Cue, change in Target, and the change in Cue*change in Target interaction (potentially with random effects of subject identity and word pair identity to combine experiments and control for pair memorability/counterbalancing), which has the additional bonus of potentially allowing the authors to include all word pairs in a single model and better describe the Osgood-style spaces in Figure 6.

This is a very interesting idea. We set this analysis up as the reviewer suggested, using fixed effects for ΔCue, ΔTarget, and ΔCue*ΔTarget, and random effects for subject and word ID. Because we had a binary outcome variable, we used mixed effects logistic regression. For a given pair, if it had the same cue or target, the corresponding change column received a 0, and if it had a different cue or target, it received a graded value (1 - GloVe value between the new and old cue or target). For this analysis, because we designed this analysis to indicate a treatment away from a repeat (as in the No Δ condition, which had no change for either cues and targets), we omitted control items. For items in the ΔBoth condition, we initially used positive values in both the Cue and Target columns too, with the multiplied ΔCue*ΔTarget value in its own column. We focused these analyses on the 48-hr delay experiments. In both experiments, running it this way resulted in highly significant negative effects of ΔCue and ΔTarget (both *p <* 0.001), but positive effects of ΔCue*ΔTarget (*p* < 0.001), presumably because after accounting for the negative independent predictions of both ΔCue and ΔTarget, ΔCue*ΔTarget values actually were better than expected.

We thought that those results were a little strange given that generally there did not appear to be interactions with ΔCue*ΔTarget values, and the positive result was simply due to the other predictors in the model. To show that this is the case, we changed the predictors so that items in the ΔBoth condition had 0 in ΔCue and ΔTarget columns alongside their ΔCue*ΔTarget value. In this case, all three factors negatively predicted memory (all *p <* 0.001).

We don't necessarily see this second approach as better, partly because it seems clear to us that any direction you go from identity is just hurting memory, and we felt the need to drop the control condition. We next flipped around the analysis to more closely resemble how we ran the other analyses, using similarity instead of distance. Here, identity along any dimension indicated a 1, a change in any part of the pair involved using that pair’s GloVe value (rather than the 1 – the GloVe value from above), and the control condition simply had zeros in all the columns. In this case, if we code the cue and target similarity values as themselves in the ΔBoth condition, in both 48-hr experiments, cue and target similarity significantly positively predicted memory (narrower set: cue similarity had *p* = 0.006, target similarity had *p* < 0.001; wider set: both *p* < 0.001) and the interaction term negatively predicted memory (*p* < 0.001 in both). If we code cue and target similarity values as 0s in the ΔBoth condition, all three factors tend to be positive (narrower, Cue: *p* = 0.11, Target and Interaction: *p* < 0.001; wider, Cue and Target *p* < 0.001; Interaction: *p* = 0.07).

Ultimately, we would prefer to leave this out of the manuscript in the interest of simplicity and because we largely find that these analyses support our prior conclusions. However, we could include them if the reviewer prefers.